# ALIGNMENT, CONVEXITY AND COMPLETENESS: MECHANISMS BEHIND GROUPDRO

## ABSTRACT

Models trained with Empirical Risk Minimization (ERM) often fail to generalize under spurious correlations. Group Robustness Methods (GRMs)—notably Group DRO (GDRO)—mitigate this by reweighting losses across groups defined by labels and spurious attributes, yet why they work remains only partially understood. We study the learning dynamics of GRMs and their effects on both the classifier head and the representation. Theoretically, in a fine-tuning setting (fixed features), we analyze the classifier learned by GDRO and show: (i) GDRO aligns less with a spurious classifier and more with an oracle non-spurious classifier than ERM; (ii) when group losses are $\mu$-strongly convex, the alignment gap controls performance, yielding an upper bound on the worst-group performance gap between ERM and GDRO; and (iii) for convex losses, adding L2 regularization induces $\mu$-strong convexity, so the same guarantees apply—providing an explanation for the empirical gains of GDRO with L2 reported in prior work. Empirically, across standard image and text benchmarks, we confirm the predicted alignment behavior. Beyond the head, under end-to-end training GDRO also reshapes the representation: through a measure called Completeness, we show that task-relevant information is spread across multiple dimensions in GDRO while ERM tends to concentrate it in fewer, making it more susceptible to rely on spurious attributes for prediction. Together, our theory and measurements clarify the mechanisms by which GDRO outperforms ERM.

## 1 INTRODUCTION

Deep learning has achieved remarkable progress in the past decade, yet models trained under the conventional Empirical Risk Minimization (ERM) paradigm (Vapnik, 1991) often fail when facing spurious correlations. For example, a model distinguishing cows from camels may rely on background cues (grass vs. beach) rather than animal shape, leading to failures under distribution shift. This phenomenon—studied under names such as *spurious correlations* (Arjovsky et al., 2020), *simplicity bias* (Shah et al., 2020), or *shortcut learning* (Geirhos et al., 2020)—appears not only in vision but also across text and other modalities (Williams et al., 2018a; Pavlopoulos et al., 2020). To address this, Group Robustness Methods (GRMs) have emerged as state-of-the-art approaches. By partitioning data into groups defined by labels and spurious attributes, they optimize for worst-group performance (Hu et al., 2018). Among them, Group Distributionally Robust Optimization (GDRO) (Sagawa* et al., 2020) has proven particularly effective, inspiring a broad line of follow-up work (Seo et al., 2022; Sohoni et al., 2020; Liu et al., 2021; Idrissi et al., 2022; Kirichenko et al., 2023). Yet, despite their empirical success, the mechanisms by which GRMs, and GDRO in particular, outperform ERM remain only partially understood. Newer methods emphasize the role of the classifier layer (e.g., Deep Feature Reweighting) (Kirichenko et al., 2023; Idrissi et al., 2022) and disregard any value in the representation learning (Bengio et al., 2014) capabilities of GDRO over ERM. These methods, however, tend to work well only when the classifier is finetuned over *unseen data*, while GDRO works well using only training data. Additionally, why GDRO requires strong L2 regularization to perform (Sagawa* et al., 2020) is still a mystery. We believe a systematic understanding of the learning dynamics of GDRO is due.

In this work, we investigate the learning dynamics of GRMs both theoretically and empirically. We focus first on the classifier, showing that GDRO induces less alignment with spurious classifiers and more with oracle non-spurious ones. We then show that this difference in spurious alignment upper bounds the difference in worst-group performance for $\mu$-strongly convex losses. Using this result,

we show that L2 regularization makes cross-entropy losses $\mu$-strongly convex—providing a mechanism by which L2 regularization is needed by GDRO. We then extend the analysis to representations, demonstrating that GDRO does not discard spurious features, but rather distributes task-relevant information across multiple dimensions, reducing reliance on any single spurious attribute. Together, these findings clarify the mechanisms underlying the success of GDRO and highlight broader principles for robust learning under spurious correlations.

**Contributions.** Our contributions can be summarized as follows:

- **Theoretical analysis of the classifier:** We show that GDRO produces classifier heads that align less with spurious classifiers and more with oracle non-spurious classifiers than ERM. For $\mu$-strongly convex losses, we prove that this alignment gap yields an upper bound on the worst-group performance difference between ERM and GDRO.

- **Role of regularization:** Using the earlier link between alignment and performance for $\mu$-strong losses, we show that GDRO needs L2 regularization on the (non-$\mu$-strong) Cross-Entropy loss because this regularization makes the loss $\mu$-strong.

- **Representation analysis:** Beyond the classifier, we show that GDRO reshapes learned representations. We find that GDRO distributes task-relevant information across multiple dimensions, making classifiers less dependent on individual spurious attributes.

- **Empirical validation:** Across standard vision and text benchmarks, we confirm the predicted alignment dynamics and representation effects, clarifying the mechanisms underlying GDRO's superior performance over ERM.

## 2 PRELIMINARIES

To begin our analysis we will first state the specific problem we will be studying and provide definitions for key concepts such as *spurious* and *non-spurious classifier*, which will be used during the rest of the article. Then, we proceed to formally define the methods we will be studying along this article which are 4: Empirical Risk Minimization (ERM), Reweighting (RW), Group Distributionally Robust Optimization (GDRO) and Subsampling (SUBG). Finally, we describe datasets and models used for our experiments.

### 2.1 PROBLEM FORMULATION

We'll work on a binary classification problem with input features $\vec{x} = \{\vec{x}_{inv}, \vec{x}_{sp}\}$, where $\vec{x}_{inv}$ represents invariant features and $\vec{x}_{sp}$ is an spurious attribute $A$ correlated with correlation $c_A$ with the ground truth label $y$. We will assume $A$ may take two values: $\{\mathcal{B}, -\mathcal{B}\}$. From $A$, we will partition our dataset into $G = 4$ groups with respect to $\vec{x}$ and $y$. Let $\mathcal{G}_A = \{s0_A, s1_A, n0_A, n1_A\}$. These are groups defined by whether the spurious correlation is predictive ("s") of the ground truth label or not ("n"), and the value of the ground truth label (0,1). Let $N_g$ be the size of group $g$ and $N = \sum_{g \in \mathcal{G}} N_g$ is the total amount of samples.

We will also assume that we will train a model on this problem using SGD, and that this model can be separated into an arbitrary non-linear feature extractor $\Phi$ and a linear classifier $C(\vec{x}) = W \cdot \vec{x} + \vec{b}$. The output of the feature extractor will be a set of features that depend on the invariant and spurious features of the data $= \Phi(\vec{x}_{inv}, \vec{x}_{sp})$. Our final model then takes the form $M(\vec{x}) = C(\Phi(\vec{x}))$

**Definition 2.1.** *Spurious Classifier. Let $\alpha = \vec{v} \cdot \vec{x}$. Define classifier $\vec{v}$ as spurious w.r.t $\mathcal{G}_A$ if:*

$$\vec{x} \in \{s0_A, n1_A\} \implies \alpha > 0, \vec{x} \in \{s1_A, n0_A\} \implies \alpha < 0$$

**Definition 2.2.** *Non-Spurious Classifier. Let $\alpha = \vec{v} \cdot \vec{x}$. Define classifier $\vec{v}$ as non-spurious w.r.t $\mathcal{G}_A$ if:*

$$\vec{x} \in \{s1_A, n1_A\} \implies \alpha > 0, \vec{x} \in \{s0_A, n0_A\} \implies \alpha < 0$$

The former definition indicates a classifier that perfectly classifies using $A$, while the latter does so according to $y$. The theoretical analyses done in Section 3 will be done according to these two definitions.

## 2.2 ROBUSTNESS METHODS

Group Robustness Methods (GRMs) are based around the idea of reweighting examples from the training set. The most successful methods like Group DRO and Reweighting assume the training distribution to be a mixture of $G$ groups. In practice, what they do is to create $G = |\mathcal{A}| \times |\mathcal{Y}|$ groups. Where $|\mathcal{A}|$ is the number of possible values an input spurious attribute might have, while $|\mathcal{Y}|$ is the number of classes in the classification problem. To define these groups requires previous knowledge of spurious correlations in the data. Plenty of methods attempt to forego this requirement for group labelling in the training phase by finding proxies for it through clustering (Seo et al., 2022) or other means (Liu et al., 2021). The most general form of these methods has the following Loss Function:

$$\mathcal{L}^{GEN} = \sum_{i=1}^{N} p_i \cdot \mathcal{L}(x_i, y_i)$$

Where $p_i$ is simply a weighting factor for the $i-$th example which can be either a constant or more complex function. Our mathematical definitions for these methods come from the actual implementations of these methods, in particular, GDRO's definition is taken from the implementation of GDRO used in JTT (Liu et al., 2021).

### 2.2.1 EMPIRICAL RISK MINIMIZATION (ERM)

This is the traditional training loss where the average loss across the training data is minimized. This method suffers heavily from spurious correlations. Its loss function becomes:

$$\mathcal{L}^{ERM} = \sum_{i=1}^{N} \frac{\mathcal{L}(x_i, y_i)}{N}$$

### 2.2.2 REWEIGHTING

This GRM (Shimodaira, 2000) involves reweighting the loss of each group in the training data, so as to mitigate the impact of each group's size. In particular, the loss function becomes simply the average loss of each group's average loss. $G$ is the number of groups in the dataset.

$$\mathcal{L}^{RW} = \frac{1}{G} \sum_{g \in \mathcal{G}} \sum_{x_i \in \mathbb{X}_g} \frac{\mathcal{L}(x_i, y_i)}{N_g}$$

### 2.2.3 GROUP DRO (GDRO)

GDRO (Sagawa* et al., 2020) tackles worst-group error by optimizing the following function ($\epsilon$ is a hyperparameter) during training, which we use for empirical evaluations:

$$\mathcal{L}^{GDRO} = \sum_{g \in \mathcal{G}} p_g \sum_{x_i \in \mathbb{X}_g} \frac{\mathcal{L}(x_i, y_i)}{N_g}, p_i = \frac{e^{\epsilon \cdot \mathcal{L}_i}}{\sum_{i=1}^{G} e^{\epsilon \cdot \mathcal{L}_i}}$$

### 2.2.4 SUBSAMPLING (SUBG-FT)

An alternate and simple baseline for inducing robustness is by using Subsampling (Idrissi et al., 2022), which consists of finetuning a model using ERM on a subsampled version of the dataset which enforces balance between groups. Usually, each group is subsampled to the size of the smallest group.

Finally, we do not include other methods such as JTT (Liu et al., 2021), DFR (Kirichenko et al., 2023), AFR (Qiu et al., 2023) or LFR(Ghaznavi et al., 2023) in our analysis because these methods in one way or another are a combination of the simpler methods studied here; they estimate weights for each sample either through proxies for group labels and/or they use Subsampling for finetuning a classifier.

**GRM implementations** Our implementation of the methods used in our experiments is mostly based off of Liu et al. (2021). For Subsampling we followed the procedure used in Kirichenko et al. (2023) but without training the 10 classifiers.

## 2.3 DATASETS

**MNIST-CIFAR (Shah et al., 2020)**   MNIST-CIFAR consists of images from MNIST (LeCun & Cortes, 2010) and CIFAR-10 (Krizhevsky et al.) concatenated vertically while a spurious correlation is induced by associating CIFAR-10 labels to MNIST labels with a tunable correlation parameter. Classes 0 and 1 from MNIST are correlated with corresponding classes of CIFAR-10. We train our models on versions of this dataset with correlations of 0.0, 0.25, 0.5, 0.75 and 0.9.

**Waterbirds (Sagawa* et al., 2020)**   Waterbirds is a dataset of real images of birds where a spurious correlation is induced with the background. The task consists of identifying if the bird is a land or sea bird, while the background is land or sea. This dataset shows 0.9 correlation between class label and spurious attribute and generate versions for correlations of 0.0, 0.25, 0.5, 0.75 and 0.9.

**CelebA (Liu et al., 2015)**   CelebA is a dataset of real images of celebrities carefully annotated with different attributes (gender, hair color, facial hair, attractiveness, etc.) which allows for extensive creation of spuriously correlated datasets. We use the same splits and attributes as in Sagawa* et al. (2020), where the target attribute is Blonde Hair and the spurious attribute is gender. We only use the original training split for this dataset, which is 0.3. However, for ease of display on tables and figures, we list all results for CelebA under correlation = 0.9.

**MultiNLI (Williams et al., 2018b)**   We use the MultiNLI corpus, a large-scale natural language inference benchmark spanning multiple domains. To study spurious correlations, we concentrate on the association between the gold label (*entailment/contradiction/neutral*) and the presence of negation words in the hypothesis (*sentence2_has_negation*).

**CivilComments (Borkan et al., 2019; Koh et al., 2020)**   We use the CivilComments-WILDS dataset, a subset of the Civil Comments platform annotated for toxicity and identity references. Following prior fairness work, we focus on spurious correlations between toxicity and the presence of demographic identity mentions (*identity_any*).

## 2.4 MODELS

For MNIST-CIFAR we use a simple convolutional network with 3 convolutional layers of 32, 64, 128 filters and a linear layer at the end, with ReLU activations. No Max-Pooling was used. For Waterbirds and CelebA we use a Resnet-50 (He et al., 2016) pretrained on ImageNet (Deng et al., 2009). For MultiNLI and CivilComments, we use the BERT architecture used by Liu et al. (2021).

## 3 ANALYSIS OF CLASSIFIER ALIGNMENT FOR ERM AND GDRO

Our theoretical analysis is carried out in the *fine-tuning setting*, where the feature extractor is frozen and only the linear classifier is trained. Under this assumption, the only way GDRO can differ from ERM is through how the final classifier aligns with different predictive directions: GDRO should align less with the spurious direction and more with an oracle non-spurious classifier, whereas ERM shows the opposite tendency. We formalize this intuition by tracking the dot product between the learned classifier and the spurious/non-spurious directions throughout training. Furthermore, we prove that when the loss is $\mu$-strongly convex, the alignment gap directly bounds the worst-group performance difference between ERM and GDRO. Since cross-entropy in this setting is convex but not $\mu$-strongly convex, we also show that adding L2 regularization induces $\mu$-strong convexity, extending our guarantees and yielding explicit bounds. We use the Min-Max formulation of GDRO for this analysis.

In this section $\theta^*$ and $\theta_{ERM}$ refer to the optimal worst group classifier and the optimal ERM classifier respectively. Our first result talks about how GDRO tends to align less with spurious classifiers and more with non-spurious ones. The full derivation of all results are in the Appendix Section B.

### 3.1 ALIGNMENT TO SPURIOUS AND NON-SPURIOUS DIRECTIONS FOR ERM AND GDRO

Our first set of results show that for even loss functions: $\alpha_{sp}(\theta^*) \leq \alpha_{sp}(\theta_{ERM})$ and $\alpha_{ns}(\theta^*) \geq \alpha_{ns}(\theta_{ERM})$ .

**Proposition (B.2 (summary)).** *Let $u = v_{sp}/\|v_{sp}\|$ and write $\theta = \theta_\perp + t\, u$, $t = \langle u, \theta \rangle$. Assume $L_g(\theta) = R(\theta_\perp) + \phi(t - a_g)$ with the same $R$ for all $g$, where $\phi \in C^1(\mathbb{R})$ is even, strictly convex, $\phi(0) = 0$, $\phi'$ odd, and $\phi$ strictly increasing on $[0, \infty)$. Let $a_- := \min_g a_g < 0 < \max_g a_g =: a_+$ with $a_+ = -a_- > 0$. Define $\theta_{\mathrm{ERM}} \in \arg\min_\theta \sum_g p_g L_g(\theta)$ and $\theta^* \in \arg\min_\theta \max_g L_g(\theta)$ and set $\alpha_{sp}(\theta) := |\langle u, \theta \rangle|$. If $P_\pm(a) := \sum_{g:\, a_g = \pm a} p_g$ :*

$$\sum_{a > 0} \big( P_-(a) - P_+(a) \big) \phi'(a) \neq 0 \implies \alpha_{sp}(\theta^*) < \alpha_{sp}(\theta_{\mathrm{ERM}})$$

**Proposition (B.3 (summary)).** *Let $u$ be a non-spurious direction. Write $\theta = \theta_\perp + t\, u$ with $t = \langle u, \theta \rangle$ and $\theta_\perp \in \{u\}^\perp$. Assume $L_g(\theta) = R(\theta_\perp) + \phi(t - a_g)$, where $R$ is the same convex function for all groups, and $\phi \in C^1(\mathbb{R})$ is even, strictly convex, satisfies $\phi(0) = 0$, and has odd, strictly increasing derivative $\phi'$. Suppose that*

$$0 < a_{\min} := \min_g a_g \ \leq a_g \leq\ a_{\max} := \max_g a_g,$$

*and let $p_g > 0$ with $\sum_g p_g = 1$. Define the ERM and GroupDRO optimizers $t_{\mathrm{ERM}} \in \arg\min_{t \in \mathbb{R}} \sum_g p_g \phi(t - a_g)$ and $t^* \in \arg\min_{t \in \mathbb{R}} \max_g \phi(t - a_g)$, and set $\alpha_{\mathrm{ns}}(\theta) := |\langle u, \theta \rangle| = |t|$. Then:*

1. *Both ERM and GroupDRO choose the same $\theta_\perp^\star \in \arg\min_{\theta_\perp} R(\theta_\perp)$, so the comparison reduces to the one-dimensional problems in $t$ above.*

2. *Let $m := \frac{1}{2}(a_{\min} + a_{\max})$. Then $t^* = m$ and $t_{\mathrm{ERM}}$ is the unique root of*

$$f'(t) := \sum_g p_g\, \phi'(t - a_g) = 0,$$

   *with $t_{\mathrm{ERM}} \in [a_{\min}, a_{\max}]$ and $t_{\mathrm{ERM}} > 0$.*

3. *We have*

$$\alpha_{\mathrm{ns}}(\theta^*) \begin{cases} > \ \alpha_{\mathrm{ns}}(\theta_{\mathrm{ERM}}) & \text{if } f'(m) > 0, \\ = \ \alpha_{\mathrm{ns}}(\theta_{\mathrm{ERM}}) & \text{if } f'(m) = 0, \\ < \ \alpha_{\mathrm{ns}}(\theta_{\mathrm{ERM}}) & \text{if } f'(m) < 0. \end{cases}$$

   *In particular, when $f'(m) > 0$, GroupDRO pushes further towards $|t^*| > |t_{\mathrm{ERM}}|$.*

In the Appendix, Propositions B.4 and B.5 show something stronger: that the relation between alignments holds also along the optimization trajectory. Our next result shows how the difference in alignment relates to the difference in worst-group performance.

## 3.2 RELATION BETWEEN ALIGNMENT, PERFORMANCE AND REGULARIZATION FOR GDRO

We assume losses are $\mu$-strongly convex, which enables relating classifier alignment to the achieved losses.

**Proposition (B.6 (summary)).** *Assume each $L_g$ is $\mu$-strongly convex. Then $L_g(\theta) \geq L_g(\theta_{ERM}) + \frac{\mu}{2}\|\theta - \theta_{ERM}\|^2$, and consequently $\alpha_{sp}(\theta_{ERM}) - \alpha_{sp}(\theta^*) \geq \sqrt{\frac{2\big(L_{\max}(\theta_{ERM}) - L_{\max}(\theta^*)\big)}{\mu}}$.*

This implies that for GDRO to attain lower loss than ERM, its alignment with the spurious direction must be smaller. Our previous results confirm this behavior, highlighting alignment as a key mechanism behind GDRO's performance gains. A limitation is that Cross-Entropy is not $\mu$-strongly convex. Empirically, GDRO requires L2 regularization (Sagawa* et al., 2020), but a theoretical justification was missing. We show that adding an L2 penalty of weight $\frac{\lambda}{2}$ yields $\lambda$-strong convexity, allowing us to connect $\lambda$, alignments and losses via a lower bound on the alignment gap.

**Proposition (B.7 (summary)).** *Let each group risk $L_g \colon \Theta \to \mathbb{R}$ be convex, and fix a spurious unit vector $u = v_{sp}/\|v_{sp}\|$. For a regularization parameter $\lambda > 0$, define $R_\lambda(\theta) = \frac{\lambda}{2}\|\theta\|^2$, $\tilde{L}_g(\theta) = L_g(\theta) + R_\lambda(\theta)$, and write $\theta_{ERM}^\lambda = \arg\min_\theta \sum_{g=1}^G p_g \tilde{L}_g(\theta)$, $\theta^{\lambda,*} = \arg\min_\theta \max_g \tilde{L}_g(\theta)$, with*

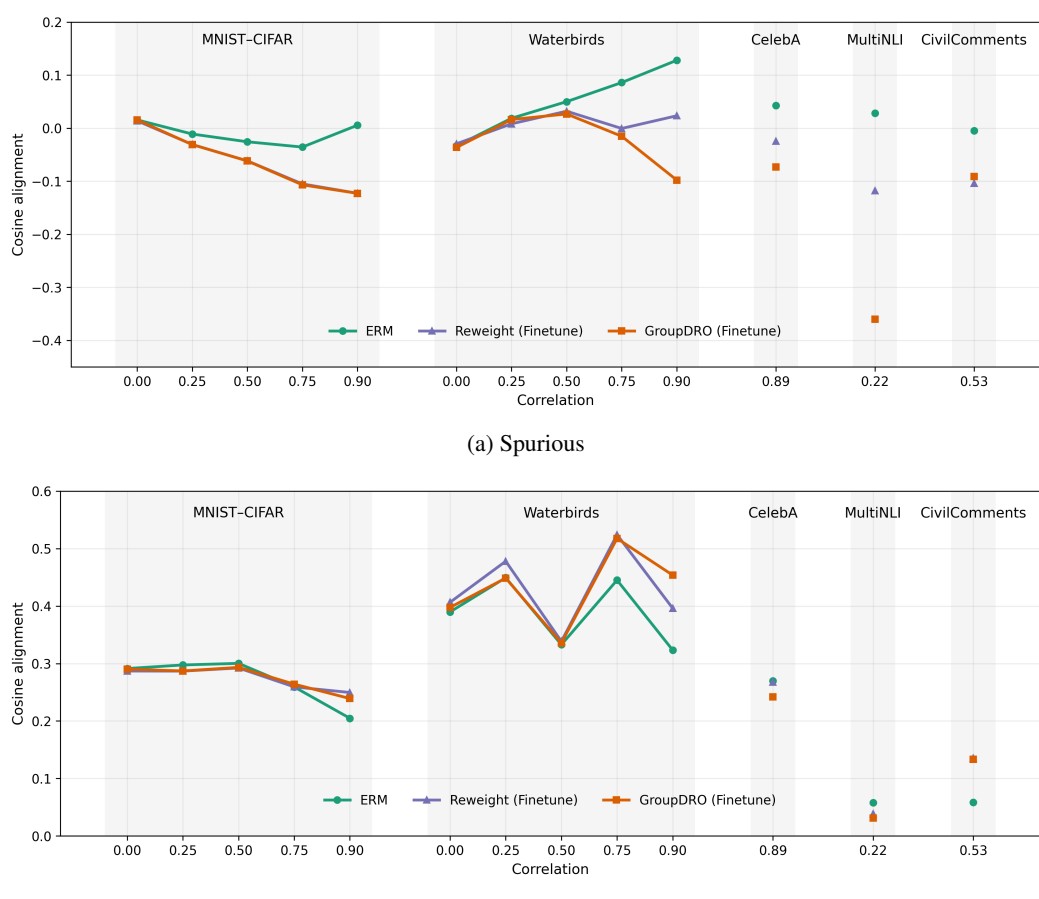

Figure 1: Cosine similarity between classifiers learned by models fine-tuned with different methods on ERM features and two reference classifiers: one predicting the spurious attribute and one oracle non-spurious classifier, across multiple datasets. (a) Consistent with our theory, GDRO exhibits lower similarity to the spurious classifier, in some cases even aligning in the opposite direction. (b) For the non-spurious classifier, GDRO is generally more aligned than ERM, except in CelebA and MultiNLI. Overall, the dominant effect is GDRO's reduced alignment with the spurious classifier.

$$\alpha_{sp}(\theta) = |u^T \theta|, \quad \tilde{L}_{\max}(\theta) = \max_g \tilde{L}_g(\theta). \text{ Then:}$$

$$\frac{\lambda(\alpha_{sp}(\theta_{ERM}^\lambda) - \alpha_{sp}(\theta^{\lambda,*}))^2}{2} \geq \tilde{L}_{\max}(\theta_{ERM}^\lambda) - \tilde{L}_{\max}(\theta^{\lambda,*})$$

We further show in Proposition B.8 in the Appendix that the result extends to quadratic regularizers of the form $R_M(\theta) = \frac{1}{2}\theta^T M \theta$ with $M$ positive definite matrix. In the next section, we show whether this alignment effect actually occurs in practice and how important it is.

## 4   EMPIRICAL EVALUATION

We first validate our theoretical alignment results empirically across multiple datasets and three methods: ERM, RW, and GDRO. To approximate the spurious classifier, we train a linear model on pretrained features to predict the spurious attribute. For the non-spurious classifier, we approximate an oracle by training on both train and test distributions to predict the class label. Results are shown in Figure 1. For the spurious classifier, both RW and GDRO consistently display lower alignment than ERM, with GDRO almost always showing the least alignment (the only exception being CivilComments, where RW is slightly lower). For the non-spurious classifier, GDRO and RW tend to be more aligned, except in CelebA and MultiNLI. We attribute these exceptions to the fact that the oracle non-spurious classifier is only well-defined in linearly separable settings, which is

Table 1: Worst Group Accuracy (%) across five benchmarks at varying levels of spurious correlation. We compare (i) end-to-end ERM, RW, and GDRO, (ii) finetuned models over ERM features (RW-FT, GDRO-FT, SUBG), and (iii) ablations combining SUBG-FT with RW or GDRO losses (RW+, GDRO+). GDRO-FT generally matches or exceeds end-to-end GDRO, RW-FT is less consistent, and SUBG-FT reliably improves over ERM. GDRO+ provides consistent gains over SUBG across all datasets. The classifier effect thus explains much of GDRO's success, but in MNIST-CIFAR and Waterbirds it falls short, suggesting that representation-level factors also play a crucial role. MC=MNIST-CIFAR, WB=Waterbirds, CA=CelebA, MNLI=MultiNLI, CC=CivilComments.

| DS | CORR | END-TO-END | | | FINETUNED | | | SUBG+DRO | |
|----|------|------|------|------|------|------|------|------|------|
| | | ERM | RW | GDRO | RW-FT | GDRO-FT | SUBG-FT | RW+ | GDRO+ |
| MC | 0.25 | 82.11% (0.4) | 86.72% (2.2) | 86.59% (2.0) | 86.86% (1.3) | 86.72% (1.2) | 87.95% (0.7) | **88.22%** (0.5) | **88.22%** (0.6) |
| | 0.5 | 78.08% (0.7) | 88.50% (0.7) | 88.34% (0.6) | **88.85%** (1.0) | 88.72% (0.8) | 87.15% (1.2) | 87.42% (0.9) | 87.01% (2.1) |
| | 0.75 | 54.86% (5.0) | 84.26% (1.1) | 83.52% (0.9) | 84.38% (0.2) | 84.13% (0.6) | 83.00% (3.0) | 84.34% (0.4) | **85.54%** (0.0) |
| | 0.9 | 29.91% (3.2) | **84.72%** (1.1) | 82.96% (0.7) | 81.70% (1.0) | 77.95% (2.0) | 79.83% (1.6) | 78.71% (1.8) | 80.59% (0.6) |
| WB | 0.25 | 89.56% (0.7) | 89.87% (1.5) | 89.50% (3.1) | 86.45% (0.4) | 86.45% (0.8) | 87.33% (0.7) | 88.32% (1.8) | **90.58%** (0.6) |
| | 0.5 | 87.44% (1.0) | 88.84% (1.0) | **90.13%** (0.5) | 82.71% (0.2) | 84.48% (0.4) | 87.80% (0.5) | 85.15% (2.7) | **90.13%** (0.7) |
| | 0.75 | 80.74% (0.8) | 88.06% (1.6) | 87.59% (0.4) | 74.35% (1.1) | 78.50% (1.2) | 84.58% (1.6) | 81.25% (2.2) | **88.32%** (0.6) |
| | 0.9 | 71.96% (0.8) | **86.31%** (0.3) | 85.87% (0.9) | 65.21% (1.9) | 73.68% (3.1) | 80.54% (1.7) | 70.63% (1.2) | 82.83% (2.0) |
| CA | 0.89 | 46.48% (1.2) | 86.48% (2.3) | 88.52% (0.3) | 86.48% (0.9) | **90.37%** (0.3) | 85.00% (1.5) | 85.56% (0.6) | 89.71% (0.3) |
| MN | 0.22 | 69.45% (2.2) | 67.42% (3.6) | 76.13% (1.2) | 71.04% (0.3) | **76.83%** (1.4) | 68.73% (1.3) | 68.99% (1.0) | 70.24% (2.3) |
| CC | 0.53 | 66.14% (2.4) | 80.00% (2.4) | 79.89% (3.3) | 81.51% (2.2) | **81.87%** (1.0) | 77.46% (6.0) | 76.92% (5.8) | 77.86% (5.1) |

not the case here, making the approximation imperfect. Overall, the results suggest that the primary driver of performance differences is the reduced alignment with the spurious classifier.

## 4.1 EMPIRICAL EFFECTS OF FINETUNING FINAL CLASSIFIER LAYER ON GRMs

A natural next step is to assess how substantial the classifier effect really is. If robustness gains can be largely attributed to the final linear classifier, as suggested by recent work (Kirichenko et al., 2023), then group-based feature learning may not be essential. To test this, we compare three groups of models: (i) end-to-end models trained with ERM, RW, and GDRO; (ii) models finetuned over ERM features (RW-FT, GDRO-FT, and Subsampling—SUBG); and (iii) ablations that combine SUBG with RW or GDRO losses (RW+, GDRO+). Results are summarized in Table 1.

Overall, finetuning using GDRO consistently yields stronger classifiers: in four out of five datasets, it markedly outperforms ERM and in three cases it actually outperforms end-to-end GDRO. RW exhibits similar tendencies but with much higher variance, sometimes surpassing ERM and sometimes falling well below it. SUBG also reliably improves over ERM, and in MNIST-CIFAR and Waterbirds it even outperforms GDRO-FT. In these same two datasets, however, end-to-end GDRO remains superior to GDRO-FT and SUBG. Remarkably, augmenting SUBG with the GDRO loss (GDRO+) consistently improves performance over SUBG alone across all datasets. We suggest using GDRO+ as a better baseline for robustness methods as it is basically a free improvement over SUBG. These findings show that the classifier effect of GDRO is indeed powerful, but not sufficient to explain all robustness gains. In particular, MNIST-CIFAR and Waterbirds highlight that end-to-end GDRO provides additional benefits, pointing to representation-level effects that we study next.

## 4.2 PROPERTIES OF GDRO FEATURES

To understand why features learned with GDRO may confer greater robustness, we consider two hypotheses: (1) GDRO does not learn spurious information during training, and (2) GDRO produces more disentangled features, making it easier for the classifier to rely on predictive rather than spurious attributes. The first hypothesis, analyzed in the Appendix in Section C, is false: GDRO learns spurious features just as ERM does. We therefore focus on the second hypothesis through a quantitative study of disentanglement. For this, we adopt the DCI framework (Eastwood & Williams, 2018), which evaluates representations along three dimensions: *Disentanglement*, measuring how specific each latent dimension is to a single attribute (0 = fully mixed, 1 = perfectly specific); *Com-*

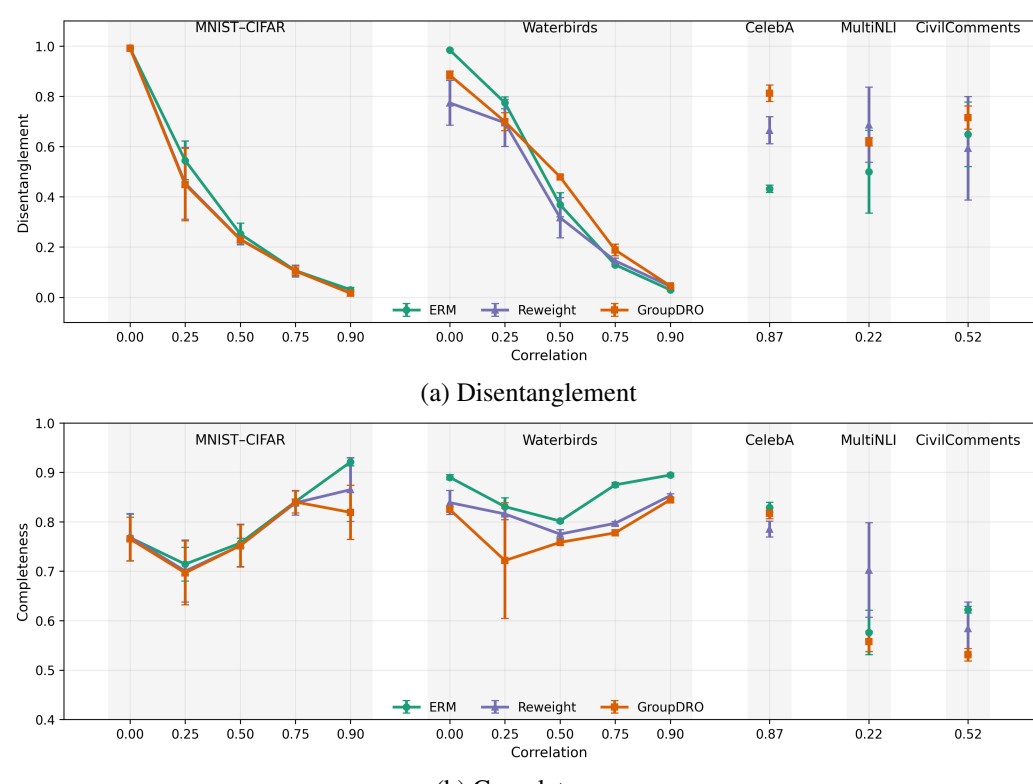

(a) Disentanglement

(b) Completeness

Figure 2: Disentanglement and Completeness of features learned with ERM, Reweighting, and GDRO across several datasets. (a) GDRO frequently increases *disentanglement*, though the effect is not uniform across datasets. (b) In contrast, *completeness* is consistently lower for GDRO, indicating that its robustness partly stems from distributing predictive information across multiple dimensions, thereby reducing reliance on spurious attributes.

*pleteness*, measuring how concentrated each attribute is across dimensions (0 = spread evenly, 1 = concentrated in one dimension); and *Informativeness*, quantifying overall predictive power, typically assessed with linear probes. In theory, a desirable representation should score high on all three. Our results, however, challenge this view in the presence of spurious correlations.

For efficiency, we reduce representations to $K = 50$ dimensions via PCA, retaining most of the variance. Figure 2 reports results across all datasets. For full numerical results, including informativeness, see Section D.3 in the Appendix. As expected, disentanglement decreases as spurious correlation increases. Unexpectedly, models trained with GDRO show higher disentanglement in more challenging datasets (e.g., CelebA, MultiNLI, CivilComments, high-correlation Waterbirds), but not in simpler cases such as MNIST-CIFAR or low-correlation Waterbirds. Completeness shows a clearer pattern: GRMs consistently achieve lower completeness than ERM across datasets. This indicates that GRM representations distribute predictive information across more dimensions, reducing reliance on any single (potentially spurious) dimension. To the best of our knowledge, this is the first time an empirical link between completeness and robustness has been established. Moreover, new robustness methods may choose to target this property to achieve their goals.

## 5 RELATED WORK

### 5.1 METHODS FOR ROBUST LEARNING

Invariant Risk Minimization (Arjovsky et al., 2020) modifies the loss function to make the model invariant to different environments. Group Reweighting schemes manipulate the importance of samples during training: Reweighting (Shimodaira, 2000) reweights group losses to eliminate the impact of group size on the loss, Group Distributionally Robust Optimization (GDRO) (Sagawa* et al., 2020) reweights group losses based on their magnitude, while other methods (Seo et al., 2022; So-

honi et al., 2020) try to apply GDRO by finding proxies for the group labels needed. Some have used biased models to reweight losses to train a debiased model(Nam et al., 2020) or used biased models' representations to train an invariant classifier(Wald et al., 2023). Other methods are based on multiple training passes: some finetune an ERM trained model on balanced data (Kirichenko et al., 2023; Qiu et al., 2023; Ghaznavi et al., 2023) or use that same model to find samples to up-weight to train a new model from scratch (Liu et al., 2021) or finetune a classifier. Others start from a balanced dataset and progressively expand the dataset during training (Deng et al., 2023). Others have worked on creating non-linear classifiers that are orthogonal to a set of attributes (Xu et al., 2022), this requires having a notion of both the train and test distribution of those attributes. Other methods have used contrastive losses (Zhang et al., 2022) to create representations that are invariant to spurious attributes. Finally, some have used Self Supervised pretrained models to estimate a logit adjustment term on the loss (Tsirigotis et al., 2023) and others have proposed adversarial training (Setlur et al., 2023) focusing on features rather than groups.

## 5.2 STUDIES ON ROBUSTNESS METHODS

Several studies highlight why robust methods are needed: networks are highly prone to simplicity bias (Shah et al., 2020), and spurious features are often simpler than core ones (Vasudeva et al., 2024). The most relevant works for us analyze representations from ERM models (Kirichenko et al., 2023; Izmailov et al., 2022), arguing that GRMs mainly affect the final classifier layer and that ERM representations suffice. Our results challenge this view: while the classifier effect is central, GRMs also shape representations in ways ERM does not. Unlike prior empirical-only studies, we provide both theoretical and empirical evidence. Other analyses claim generalized reweighting offers no advantage over ERM (Zhai et al., 2023), though under restrictive assumptions. In contrast, our analysis only assumes a linear classifier over frozen features. Recent theory has studied spurious feature memorization in random/NTK settings (Bombari & Mondelli, 2024), introducing alignment measures across sample pairs. Our alignment notion differs: we quantify alignment between the learned classifier and reference spurious/non-spurious classifiers. Additional works compare methods such as GDRO in systematic generalization settings (Ahmed et al., 2021), but without addressing underlying mechanisms. Others link mutual information between spurious attributes and labels to worst-group error (Zhang et al., 2022), also invoking alignment, though defined in representation space between samples while we focus on classifier-level alignment.

## 6 CONCLUSIONS

We studied why Group Robustness Methods, and GDRO in particular, succeed where ERM fails under spurious correlations. Our theoretical analysis in the fine-tuning setting showed that GDRO produces classifiers less aligned with spurious directions and more aligned with oracle non-spurious ones, and that this alignment gap explains improvements in worst-group performance when the loss is $\mu$-strongly convex. Moreover, we proved that L2 regularization induces $\mu$-strong convexity in cross-entropy, providing a principled explanation for the necessity of strong regularization in GDRO. Empirically, we confirmed these predictions across vision and text benchmarks: GDRO reduces alignment with spurious classifiers and increases alignment with non-spurious ones compared to ERM. Going beyond the classifier, we showed that GDRO also reshapes the learned representations. Contrary to the intuition that robustness comes from discarding spurious features, we found that GDRO representations still encode them but distribute predictive information across more dimensions (lower completeness), making classifiers less dependent on individual spurious attributes. Taken together, our findings clarify the mechanisms by which GRMs achieve robustness: they act both at the classifier level and at the representation level, with effects that complement each other. We believe these insights can inspire the design of new methods that combine alignment and lower completeness, ultimately leading to more principled and practical approaches for robust learning under spurious correlations.

## 7 REPRODUCIBILITY STATEMENT

We share our code for our experiments, which is modified from Liu et al. (2021), in the supplementary materials. This includes the Jupyter Notebooks used to obtain plots and tables as well as sample scripts to reproduce results. Proof for results on the theoretical section are found in the Appendix, Section B. MNIST-CIFAR, CelebA, Waterbirds, MultiNLI and CivilComments are freely available datasets. Hyperparameters used are detailed in Section F.1.

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

## A    DATASET EXAMPLES

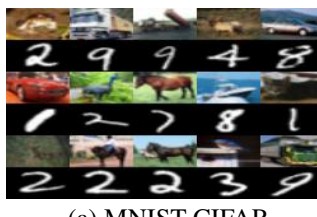 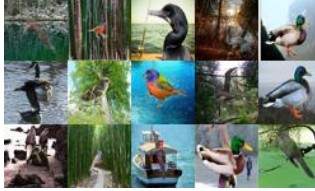 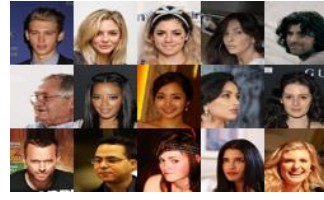

(a) MNIST-CIFAR                (b) Waterbirds                (c) CelebA

Figure 3: Sample images from datasets used in this work. MNIST-CIFAR correlates MNIST digits with CIFAR-10 classes; Waterbirds correlates land/water birds with a land/water background; CelebA consists of heavily annotated images from celebrities. In this work, we use the correlation that appears between gender and hair color.

## B    DERIVATION OF THEORETICAL RESULTS

### B.1    PRELIMINARIES

**Theorem B.1.** *Let $\mathcal{G} = \{1, \ldots G\}$ be a finite set of groups. Consider $\mathcal{D}_g$ the set of the training points in g with $|\mathcal{D}_g| = n_g > 0$. Also, denote $p_g = n_g/n$ and $n = \sum_g n_g$. Let $\ell \colon \Theta \times \mathcal{X} \times \mathcal{Y} \to \mathbb{R}$ be a proper, convex and lower bounded loss function in $\theta \in \Theta \subset \mathbb{R}^d$. Where $\Theta$ is a closed convex set. Define the group risk as*

$$L_g(\theta) = \frac{1}{n_g} \sum_{(x,y) \in \mathcal{D}_g} \ell(\theta; x, y)$$

*and the simplex $\Delta_G = \left\{ w \in \mathbb{R}_{\geq 0}^G \colon \sum_{g=1}^G w_g = 1 \right\}$. Consider the following:*

*1. $F(w) = \inf\limits_{\theta \in \Theta} \sum\limits_{g=1}^G w_g L_g(\theta)$*

*2. $\theta_{ERM} = \underset{\theta}{\operatorname{argmin}} \sum\limits_g p_g L_g(\theta)$*

*3. $\theta^* = \underset{\theta}{\operatorname{argmin}} \max\limits_g L_g(\theta)$.*

*Then*

*1. the function $F \colon \Delta_G \to \mathbb{R} \cup \{-\infty\}$ is concave,*

*2. $\min\limits_{\theta \in \Theta} \max\limits_g L_g(\theta) = \max\limits_{w \in \Delta_g} F(w)$*

*3. if $L_{\max}(\theta) = \max\limits_g L_g(\theta)$, then $L_{\max}(\theta^*) \leq L_{\max}(\theta_{ERM})$.*

*Proof.* Let's start with a proposition.

**Proposition B.1.** *The simplex $\Delta_G$ with the standard Euclidean topology is a convex, compact subset of $\mathbb{R}^G$.*

*Proof.* Consider $u, v \in \Delta_G$ and $\lambda \in [0, 1]$. Define

$$w = \lambda u + (1 - \lambda)v = \big(\lambda u_1 + (1 - \lambda)v_1, \ldots, \lambda u_G + (1 - \lambda)v_G\big).$$

Then for each $g$, we have

$$
\begin{aligned}
w_g &= \lambda u_g + (1-\lambda)v_g \\
&\geq \lambda \cdot 0 + (1-\lambda) \cdot 0 \\
&= 0,
\end{aligned}
$$

$$
\begin{aligned}
\sum_{g=1}^{G} w_g &= \sum_g \big[\lambda u_g + (1-\lambda)v_g\big] \\
&= \lambda \sum_g u_g + (1-\lambda) \sum_g v_g \\
&= \lambda \cdot 1 + (1-\lambda) \cdot 1 \\
&= 1.
\end{aligned}
$$

Hence $w \in \Delta_G$, and thus $\Delta_G$ is convex.

Now let

$$
\Delta_G = \bigcap_{g=1}^{G}\{w \in \mathbb{R}^G : w_g \geq 0\} \cap \Big\{w : \sum_g w_g = 1\Big\}.
$$

Each half-space $\{w : w_g \geq 0\}$ is closed in $\mathbb{R}^G$, and the hyperplane $\{\sum_g w_g = 1\}$ is the preimage of the closed set $\{1\}$ under the continuous map $w \mapsto \sum_g w_g$. A finite intersection of closed sets is closed, so $\Delta_G$ is closed.

Finally, for every $w \in \Delta_G$ and each coordinate $g$,

$$
0 \leq w_g \leq \sum_{h=1}^{G} w_h = 1
$$

so $\Delta_G \subset [0,1]^G$, which is bounded in $\mathbb{R}^G$. Thus, since any closed and bounded subset of $\mathbb{R}^G$ is compact, we have that $\Delta_G$ is compact. $\qquad\square$

Let's go back to the proof of Theorem B.1. By the convexity, continuity and coercivity assumptions each of the objectives

$$
\theta \mapsto \sum_g p_g L_g(\theta), \qquad \theta \mapsto \max_g L_g(\theta),
$$

admits a minimizer on the closed convex set $\Theta$.

Fix $w^{(1)}, w^{(2)} \in \Delta_G$ and $\lambda \in [0,1]$. Since for each $\theta$ the map $w \mapsto \sum_g w_g L_g(\theta)$ is affine, we have

$$
\begin{aligned}
F\big(\lambda w^{(1)} + (1-\lambda)w^{(2)}\big) &= \inf_\theta \big[\lambda f(\theta, w^{(1)}) + (1-\lambda)f(\theta, w^{(2)})\big] \\
&\geq \lambda F(w^{(1)}) + (1-\lambda)F(w^{(2)}),
\end{aligned}
$$

where $f(\theta, w) = \sum_g w_g L_g(\theta)$ and we used $\inf(f_1 + f_2) \geq \inf f_1 + \inf f_2$. Hence $F$ is concave and, being an infimum of continuous functions, upper-semi-continuous on the compact simplex $\Delta_G$. Therefore $\max_{w \in \Delta_G} F(w)$ is attained at some $w^*$.

Now define the saddle function

$$
\Phi(\theta, w) = \sum_{g=1}^{G} w_g L_g(\theta).
$$

For fixed $w$, $\Phi(\cdot, w)$ is convex and lower-bounded; for fixed $\theta$, $\Phi(\theta, \cdot)$ is affine (hence concave). Then we have

$$
\min_\theta \max_{w \in \Delta_G} \Phi(\theta, w) = \max_{w \in \Delta_G} \min_\theta \Phi(\theta, w).
$$

Since $\max_{w \in \Delta_G} \Phi(\theta, w) = \max_g L_g(\theta)$ and $\min_\theta \Phi(\theta, w) = F(w)$, we obtain

$$
\min_\theta \max_g L_g(\theta) = \max_{w \in \Delta_G} F(w).
$$

If $w^*$ attains the maximum, then any minimizer $\theta^*$ in $\min_\theta \Phi(\theta, w^*)$ also minimizes the worst-group risk, establishing the claimed equivalence.

Observe that for any $w \in \Delta_G$ and any $\theta$, $\max_g L_g(\theta) \geq \sum_{g=1}^G w_g L_g(\theta)$. Taking $w = w^*$ and applying first at $\theta_{\mathrm{ERM}}$ and then at $\theta^*$, and using optimality of $\theta^*$ for weights $w^*$, yields

$$\max_g L_g(\theta_{\mathrm{ERM}}) \geq \sum_g w_g^* L_g(\theta_{\mathrm{ERM}})$$

$$\geq \sum_g w_g^* L_g(\theta^*)$$

$$= \max_g L_g(\theta^*).$$

If the group risks at $\theta_{\mathrm{ERM}}$ are not all equal, the first inequality is strict, completing the proof. $\square$

### B.2 ALIGNMENT TO SPURIOUS AND NON-SPURIOUS DIRECTIONS FOR ERM AND GDRO

**Proposition B.2.** *Let* $u = v_{sp}/\|v_{sp}\|$ *and write* $\theta = \theta_\perp + t\,u$, $t = \langle u, \theta \rangle$. *Assume* $L_g(\theta) = R(\theta_\perp) + \phi(t - a_g)$ *with the same* $R$ *for all* $g$, *where* $\phi \in C^1(\mathbb{R})$ *is even, strictly convex,* $\phi(0) = 0$, $\phi'$ *odd, and* $\phi$ *strictly increasing on* $[0, \infty)$. *Let*

$$a_- := \min_g a_g < 0 < \max_g a_g =: a_+$$

*with* $a_+ = -a_- > 0$. *Define*

$$\theta_{\mathrm{ERM}} \in \arg\min_\theta \sum_g p_g L_g(\theta)$$

$$\theta^* \in \arg\min_\theta \max_g L_g(\theta),$$

*and set* $\alpha_{sp}(\theta) := |\langle u, \theta \rangle|$. *If*

$$\sum_{a>0} \big(P_-(a) - P_+(a)\big)\phi'(a) \neq 0,$$

*where* $P_\pm(a) := \sum_{g:\, a_g = \pm a} p_g$, *then*

$$\alpha_{sp}(\theta^*) < \alpha_{sp}(\theta_{\mathrm{ERM}}).$$

*Proof.* With $R$ identical, the objectives decouple in $(\theta_\perp, t)$. Both ERM and DRO share the same $\theta_\perp^\star \in \arg\min_{\theta_\perp} R(\theta_\perp)$. The ERM $t$ minimizes $f(t) = \sum_g p_g \phi(t - a_g)$; $f$ is strictly convex and $f'(0) = \sum_{a>0}(P_+(a) - P_-(a))\phi'(a) \neq 0$, hence $t_{\mathrm{ERM}} \neq 0$ and $|t_{\mathrm{ERM}}| > 0$. For DRO, we minimize $\max_g \phi(t - a_g)$. Since $\phi$ is even and strictly increasing on $[0, \infty)$, this is minimized at the Chebyshev center of $\{a_g\}$, which under the symmetric extremes assumption is $t^* = 0$. Therefore $\alpha_{sp}(\theta^*) = 0 < |t_{\mathrm{ERM}}| = \alpha_{sp}(\theta_{\mathrm{ERM}})$. $\square$

**Proposition B.3.** *Let* $u$ *be a non-spurious direction. Write* $\theta = \theta_\perp + t\,u$ *with* $t = \langle u, \theta \rangle$ *and* $\theta_\perp \in \{u\}^\perp$. *Assume* $L_g(\theta) = R(\theta_\perp) + \phi(t - a_g)$, *where* $R$ *is the same convex function for all groups, and* $\phi \in C^1(\mathbb{R})$ *is even, strictly convex, satisfies* $\phi(0) = 0$, *and has odd, strictly increasing derivative* $\phi'$. *Suppose that*

$$0 < a_{\min} := \min_g a_g \leq a_g \leq a_{\max} := \max_g a_g,$$

*and let* $p_g > 0$ *with* $\sum_g p_g = 1$. *Define the ERM and GroupDRO optimizers*

$$t_{\mathrm{ERM}} \in \arg\min_{t \in \mathbb{R}} \sum_g p_g \phi(t - a_g)$$

$$t^* \in \arg\min_{t \in \mathbb{R}} \max_g \phi(t - a_g),$$

*and set* $\alpha_{\mathrm{ns}}(\theta) := |\langle u, \theta \rangle| = |t|$.

*Then*

1. *Both ERM and DRO choose the same $\theta_\perp^\star \in \arg\min_{\theta_\perp} R(\theta_\perp)$, so the comparison reduces to the one-dimensional problems in $t$ above.*

2. *Let $m := \frac{1}{2}(a_{\min} + a_{\max})$. Then $t^* = m$ and $t_{\mathrm{ERM}}$ is the unique root of*

$$f'(t) := \sum_g p_g \, \phi'(t - a_g) = 0,$$

*with $t_{\mathrm{ERM}} \in [a_{\min}, a_{\max}]$ and $t_{\mathrm{ERM}} > 0$.*

3. *We have*

$$\alpha_{\mathrm{ns}}(\theta^*) \begin{cases} > \alpha_{\mathrm{ns}}(\theta_{\mathrm{ERM}}) & \text{if } f'(m) > 0, \\ = \alpha_{\mathrm{ns}}(\theta_{\mathrm{ERM}}) & \text{if } f'(m) = 0, \\ < \alpha_{\mathrm{ns}}(\theta_{\mathrm{ERM}}) & \text{if } f'(m) < 0. \end{cases}$$

*In particular, when $f'(m) > 0$, GroupDRO pushes further towards $|t^*| > |t_{\mathrm{ERM}}|$.*

*Proof.*      1. Since $R$ is identical across groups, both ERM and DRO minimize $R(\theta_\perp)$ independently of $t$, hence share $\theta_\perp^\star \in \arg\min R$. The problems decouple to the one-dimensional $t$–objectives.

2. Define $h(t) := \max_g \phi(t - a_g)$. Because $\phi$ is even and strictly increasing on $[0, \infty)$, $\arg\min_t h(t) = \arg\min_t \max_g |t - a_g|$. This is attained at the midrange $m = (a_{\min} + a_{\max})/2$, where the two farthest groups (at the extremes) are equidistant; moving $t$ left or right increases the maximal deviation. Thus $t^* = m$.

For ERM, $f(t) := \sum_g p_g \phi(t - a_g)$ is strictly convex, so $f'$ is strictly increasing and has a unique zero $t_{\mathrm{ERM}}$. Moreover,

$$f'(a_{\min}) = \sum_g p_g \, \phi'(a_{\min} - a_g) \le 0, \qquad f'(a_{\max}) = \sum_g p_g \, \phi'(a_{\max} - a_g) \ge 0,$$

so $t_{\mathrm{ERM}} \in [a_{\min}, a_{\max}]$. Since $a_g > 0$ for all $g$ and $\phi'$ is odd and increasing,

$$f'(0) = \sum_g p_g \, \phi'(-a_g) = -\sum_g p_g \, \phi'(a_g) < 0,$$

hence, by monotonicity of $f'$, the unique root satisfies $t_{\mathrm{ERM}} > 0$.

3. Because $f'$ is strictly increasing, the sign of $f'(m)$ determines the order of $t_{\mathrm{ERM}}$ relative to $m$: if $f'(m) > 0$ then $t_{\mathrm{ERM}} < m$; if $f'(m) < 0$ then $t_{\mathrm{ERM}} > m$; if $f'(m) = 0$ then $t_{\mathrm{ERM}} = m$. Since $a_{\min} > 0$, both $m$ and $t_{\mathrm{ERM}}$ are positive, so $\alpha_{\mathrm{ns}}(\theta^*) = |t^*| = m$ and $\alpha_{\mathrm{ns}}(\theta_{\mathrm{ERM}}) = |t_{\mathrm{ERM}}| = t_{\mathrm{ERM}}$, yielding the stated trichotomy.

□

**Proposition B.4.** *Let $p_g > 0$, $\sum_g p_g = 1$, and consider the continuous-time gradient flows*

$$\dot{\theta}_{\mathrm{ERM}}(t) = -\sum_{g=1}^{G} p_g \, \nabla L_g\big(\theta_{\mathrm{ERM}}(t)\big)$$

$$\dot{\theta}_{\mathrm{DRO}}(t) = -\nabla L_{k(t)}\big(\theta_{\mathrm{DRO}}(t)\big),$$

*where $k(t) \in \arg\max_g L_g\big(\theta_{\mathrm{DRO}}(t)\big)$ is any measurable selection. Fix a spurious direction $v_{sp} \in \mathbb{R}^d$ and define $\alpha_{sp}(t) := v_{sp}^\top \theta(t)$ and $\Delta\alpha_{sp}(t) := \frac{d}{dt}\alpha_{sp}(t) = v_{sp}^\top \dot{\theta}(t)$.*

*Assume the following* spurious-gradient monotonicity *at the point $\theta$:*

$$L_i(\theta) \ge L_j(\theta) \implies v_{sp}^\top \nabla L_i(\theta) \ge v_{sp}^\top \nabla L_j(\theta) \qquad \text{for all } i, j \in \{1, \dots, G\}. \tag{1}$$

*Then, at $\theta$,*

$$\Delta\alpha_{sp}\big|_{\mathrm{ERM}} \ge \Delta\alpha_{sp}\big|_{\mathrm{DRO}}.$$

*Proof.* Fix $\theta$ and abbreviate $x_g := v_{sp}^\top \nabla L_g(\theta)$. Then

$$\Delta\alpha_{sp}\big|_{\mathrm{ERM}} = v_{sp}^\top \dot\theta_{\mathrm{ERM}} = -\sum_g p_g x_g,$$

$$\Delta\alpha_{sp}\big|_{\mathrm{DRO}} = v_{sp}^\top \dot\theta_{\mathrm{DRO}} = -x_k,$$

with $k \in \arg\max_g L_g(\theta)$. Under inequality 1, any maximizer of $L_g(\theta)$ is also a maximizer of $x_g$; hence $x_k = \max_g x_g$. Since $\sum_g p_g x_g \le \max_g x_g$,

$$\Delta\alpha_{sp}\big|_{\mathrm{ERM}} = -\sum_g p_g x_g \ge -\max_g x_g = -x_k = \Delta\alpha_{sp}\big|_{\mathrm{DRO}}.$$

If not all $x_g$ on the support of $\{p_g\}$ equal $\max_g x_g$, then $\sum_g p_g x_g < \max_g x_g$, which yields strict inequality. $\qquad\square$

**Proposition B.5.** *Let $u \in \mathbb{R}^d$ be a fixed non-spurious direction (assume $\|u\| = 1$ w.l.o.g.). Consider the continuous-time gradient flows started at the same $\theta(0)$*

$$\dot\theta_{\mathrm{ERM}}(t) = -\sum_{g=1}^G p_g \nabla L_g\big(\theta_{\mathrm{ERM}}(t)\big),$$

$$\dot\theta_{\mathrm{DRO}}(t) = -\nabla L_{k(t)}\big(\theta_{\mathrm{DRO}}(t)\big),$$

*where $k(t) \in \arg\max_g L_g(\theta_{\mathrm{DRO}}(t))$ is any measurable selection and $p_g > 0$, $\sum_g p_g = 1$. Define the non-spurious projection $\alpha_{\mathrm{ns}}(t) := \langle u, \theta(t)\rangle$ and its instantaneous rate $\Delta\alpha_{\mathrm{ns}}(t) := \frac{d}{dt}\alpha_{\mathrm{ns}}(t) = \langle u, \dot\theta(t)\rangle$.*

*Fix a point $\theta$ and set $x_g := \langle u, \nabla L_g(\theta)\rangle$. Assume the following useful-gradient anti-monotonicity at $\theta$*

$$L_i(\theta) \ge L_j(\theta) \implies \langle u, \nabla L_i(\theta)\rangle \le \langle u, \nabla L_j(\theta)\rangle \quad \text{for all } i,j \in \{1,\dots,G\}. \tag{2}$$

*Then, at $\theta$,*

$$\Delta\alpha_{\mathrm{ns}}\big|_{\mathrm{ERM}} \le \Delta\alpha_{\mathrm{ns}}\big|_{\mathrm{DRO}}.$$

*Proof.* At the common point $\theta$,

$$\Delta\alpha_{\mathrm{ns}}\big|_{\mathrm{ERM}} = \Big\langle u, -\sum_g p_g \nabla L_g(\theta)\Big\rangle = -\sum_g p_g x_g,$$

$$\Delta\alpha_{\mathrm{ns}}\big|_{\mathrm{DRO}} = \langle u, -\nabla L_k(\theta)\rangle = -x_k,$$

where $k \in \arg\max_g L_g(\theta)$. By inequality 2, any maximizer of $L_g$ is a minimizer of $x_g$, hence $x_k = \min_g x_g$. Since a weighted average is lower-bounded by the minimum, $\sum_g p_g x_g \ge \min_g x_g$, so

$$\Delta\alpha_{\mathrm{ns}}\big|_{\mathrm{ERM}} = -\sum_g p_g x_g \le -\min_g x_g = \Delta\alpha_{\mathrm{ns}}\big|_{\mathrm{DRO}}.$$

Strict inequality holds if some $x_g$ is strictly larger than $\min_g x_g$ with positive weight. $\qquad\square$

## B.3   RELATION BETWEEN ALIGNMENT, PERFORMANCE AND REGULARIZATION FOR GDRO

**Proposition B.6.** *Assume each $L_g$ is $\mu$-strongly convex. Then*

$$L_g(\theta) \ge L_g(\theta_{ERM}) + \frac{\mu}{2}\|\theta - \theta_{ERM}\|^2,$$

*and consequently*

$$\alpha_{sp}(\theta_{ERM}) - \alpha_{sp}(\theta^*) \ge \sqrt{\frac{2\big(L_{\max}(\theta_{ERM}) - L_{\max}(\theta^*)\big)}{\mu}}.$$

*Proof.* By Part (2) of Theorem B.1 we have,

$$L_{\max}(\theta^*) = \min_\theta \max_g L_g(\theta) = \max_{w \in \Delta_G} \inf_\theta \sum_{g=1}^{G} w_g L_g(\theta).$$

In particular, plugging in the ERM weights $p = (p_1, \ldots, p_G)$ gives

$$L_{\max}(\theta^*) \geq \inf_\theta \sum_{g=1}^{G} p_g L_g(\theta) = \sum_{g=1}^{G} p_g L_g(\theta_{ERM}) = L_{ERM}(\theta_{ERM}),$$

where $L_{ERM}(\theta) = \sum_g p_g L_g(\theta)$. But by definition $L_{ERM}(\theta_{ERM}) \leq L_{\max}(\theta_{ERM})$.
Consequently

$$L_{\max}(\theta_{ERM}) - L_{\max}(\theta^*) \geq L_{\max}(\theta_{ERM}) - L_{ERM}(\theta_{ERM}) \geq 0.$$

Since each $L_g$ is $\mu$-strongly convex,

$$L_g(\theta_{ERM}) \geq L_g(\theta^*) + \nabla L_g(\theta^*)^\top (\theta_{ERM} - \theta^*) + \tfrac{\mu}{2} \|\theta_{ERM} - \theta^*\|^2.$$

Taking the maximum over $g$ and noting $\max_g \nabla L_g(\theta^*)^\top (\theta_{ERM} - \theta^*) \geq 0$ yields

$$L_{\max}(\theta_{ERM}) \geq L_{\max}(\theta^*) + \tfrac{\mu}{2} \|\theta_{ERM} - \theta^*\|^2.$$

Rearrange to get

$$\|\theta_{ERM} - \theta^*\| \leq \sqrt{\frac{2\big(L_{\max}(\theta_{ERM}) - L_{\max}(\theta^*)\big)}{\mu}}.$$

Finally, since $\alpha_{sp}(\theta) = |v_{sp}^\top \theta|$, the one-dimensional distance along $v_{sp}$ is at most the full Euclidean distance

$$\alpha_{sp}(\theta_{ERM}) - \alpha_{sp}(\theta^*) \geq \big\|(\theta_{ERM} - \theta^*)\big\| \geq \sqrt{\frac{2\big(L_{\max}(\theta_{ERM}) - L_{\max}(\theta^*)\big)}{\mu}}.$$

$\square$

**Proposition B.7.** *Let each group risk $L_g \colon \Theta \to \mathbb{R}$ be convex, and fix a spurious unit vector $u = v_{sp}/\|v_{sp}\|$. For a regularization parameter $\lambda > 0$, define*

$$R_\lambda(\theta) = \frac{\lambda}{2} \|\theta\|^2, \quad \tilde{L}_g(\theta) = L_g(\theta) + R_\lambda(\theta),$$

*and write*

$$\theta_{ERM}^\lambda = \arg\min_\theta \sum_{g=1}^{G} p_g \tilde{L}_g(\theta), \quad \theta^{\lambda,*} = \arg\min_\theta \max_g \tilde{L}_g(\theta),$$

*with $\alpha_{sp}(\theta) = |u^T \theta|$, $\tilde{L}_{\max}(\theta) = \max_g \tilde{L}_g(\theta)$. Then*

$$\frac{\lambda(\alpha_{sp}(\theta_{ERM}^\lambda) - \alpha_{sp}(\theta^{\lambda,*}))^2}{2} \geq \tilde{L}_{\max}(\theta_{ERM}^\lambda) - \tilde{L}_{\max}(\theta^{\lambda,*})$$

*Proof.* Since $R_\lambda(\theta)$ is $\lambda$-strongly convex and each $L_g$ is convex, each $\tilde{L}_g = L_g + R_\lambda$ is $\lambda$-strongly convex. We then proceed as follows.

By $\lambda$-strong convexity, for any $\theta$ and any group $g$,

$$\tilde{L}_g(\theta) \geq \tilde{L}_g(\theta_{ERM}^\lambda) + \nabla \tilde{L}_g(\theta_{ERM}^\lambda)^T (\theta - \theta_{ERM}^\lambda) + \tfrac{\lambda}{2} \|\theta - \theta_{ERM}^\lambda\|^2.$$

Taking maximum over $g$ and using that $\max_g \nabla \tilde{L}_g(\theta_{ERM}^\lambda)^T (\theta - \theta_{ERM}^\lambda) \geq 0$ (since $\sum_g p_g \nabla \tilde{L}_g(\theta_{ERM}^\lambda) = 0$) yields

$$\tilde{L}_{\max}(\theta) \geq \tilde{L}_{\max}(\theta_{ERM}^\lambda) + \tfrac{\lambda}{2} \|\theta - \theta_{ERM}^\lambda\|^2.$$

By the minimax equality,

$$\tilde{L}_{\max}(\theta^{\lambda,*}) = \min_\theta \max_g \tilde{L}_g(\theta) = \max_{w \in \Delta_G} \inf_\theta \sum_g w_g \tilde{L}_g(\theta) \geq \inf_\theta \sum_g p_g \tilde{L}_g(\theta) = \sum_g p_g \tilde{L}_g(\theta^\lambda_{ERM}),$$

so in particular $\tilde{L}_{\max}(\theta^\lambda_{ERM}) - \tilde{L}_{\max}(\theta^{\lambda,*}) \geq 0$.

Combining with the quadratic growth at $\theta = \theta^{\lambda,*}$ gives

$$\tfrac{\lambda}{2} \left\| \theta^{\lambda,*} - \theta^\lambda_{ERM} \right\|^2 \leq \tilde{L}_{\max}(\theta^\lambda_{ERM}) - \tilde{L}_{\max}(\theta^{\lambda,*}),$$

hence

$$\left\| \theta^\lambda_{ERM} - \theta^{\lambda,*} \right\| \leq \sqrt{\frac{2 \left( \tilde{L}_{\max}(\theta^\lambda_{ERM}) - \tilde{L}_{\max}(\theta^{\lambda,*}) \right)}{\lambda}}.$$

Finally, since $\alpha_{sp}(\theta) = |u^T \theta| \leq \|\theta\|$, we have

$$\alpha_{sp}(\theta^\lambda_{ERM}) - \alpha_{sp}(\theta^{\lambda,*}) = |u^T(\theta^\lambda_{ERM} - \theta^{\lambda,*})| \geq \|\theta^\lambda_{ERM} - \theta^{\lambda,*}\| \geq \sqrt{\frac{2 \left( \tilde{L}_{\max}(\theta^\lambda_{ERM}) - \tilde{L}_{\max}(\theta^{\lambda,*}) \right)}{\lambda}}.$$

This completes the proof under $\ell_2$ regularization. $\qquad\square$

**Proposition B.8.** *Assume each group risk $L_g : \Theta \to \mathbb{R}$ is convex and continuously differentiable. Let $M \in \mathbb{R}^{d \times d}$ be symmetric positive definite, and define the regularizer*

$$R_M(\theta) = \tfrac{1}{2} \theta^T M \theta.$$

*Fix any unit vector $w \in \mathbb{R}^d$, and write*

$$\alpha_w(\theta) = \left| w^T \theta \right|.$$

*For a parameter set of probabilities $\{p_g\}$, define*

$$\tilde{L}_g(\theta) = L_g(\theta) + R_M(\theta), \quad \theta_{ERM} = \arg\min_\theta \sum_{g=1}^G p_g \tilde{L}_g(\theta), \quad \theta^* = \arg\min_\theta \max_g \tilde{L}_g(\theta),$$

*and set*

$$\Delta L = \tilde{L}_{\max}(\theta_{ERM}) - \tilde{L}_{\max}(\theta^*), \quad \tilde{L}_{\max}(\theta) = \max_g \tilde{L}_g(\theta).$$

*Let $\mu_w = w^T M w > 0$. Then the following directional quadratic bound holds:*

$$\tfrac{\mu_w}{2} \left( \alpha_w(\theta_{ERM}) - \alpha_w(\theta^*) \right)^2 \leq \Delta L.$$

*Proof.* Since $M \succ 0$, the function $\theta \mapsto R_M(\theta)$ is $\mu_w$-strongly convex along $w$. In particular, for each $g$ and any $\theta$ the strong convexity inequality in direction $w$ gives

$$R_M(\theta) \geq R_M(\theta_{ERM}) + \nabla R_M(\theta_{ERM})^T (\theta - \theta_{ERM}) + \tfrac{\mu_w}{2} \left( w^T \theta - w^T \theta_{ERM} \right)^2.$$

Adding the convex term $L_g$ preserves this inequality:

$$\tilde{L}_g(\theta) \geq \tilde{L}_g(\theta_{ERM}) + \nabla \tilde{L}_g(\theta_{ERM})^T (\theta - \theta_{ERM}) + \tfrac{\mu_w}{2} \left( w^T \theta - w^T \theta_{ERM} \right)^2.$$

Set $\theta = \theta^*$ and take the maximum over $g$. Using the minimax identity

$$\tilde{L}_{\max}(\theta^*) = \min_\theta \max_g \tilde{L}_g(\theta) = \max_{v \in \Delta_G} \inf_\theta \sum_g v_g \tilde{L}_g(\theta) \geq \sum_g p_g \tilde{L}_g(\theta_{ERM}),$$

we obtain

$$\tilde{L}_{\max}(\theta_{ERM}) - \tilde{L}_{\max}(\theta^*) \geq \sum_g p_g \left[ \nabla \tilde{L}_g(\theta_{ERM})^T (\theta^* - \theta_{ERM}) + \tfrac{\mu_w}{2} \left( w^T \theta^* - w^T \theta_{ERM} \right)^2 \right].$$

Stationarity of $\theta_{ERM}$ implies $\sum_g p_g \nabla \tilde{L}_g(\theta_{ERM}) = 0$, so the linear term vanishes. Hence

$$\Delta L \geq \tfrac{\mu_w}{2} \left( w^T \theta^* - w^T \theta_{ERM} \right)^2 = \tfrac{\mu_w}{2} \left( \alpha_w(\theta_{ERM}) - \alpha_w(\theta^*) \right)^2.$$

This completes the proof. $\qquad\square$

## C  DO GRMS ELIMINATE INFORMATION ABOUT SPURIOUS FEATURES?

We will analyze first if the representations learned by GRMs contain information about spurious features. One hypothesis about their success might relate to them discarding this information during training. To do this, we obtain the singular vectors of a PCA decomposition of representations of the training set. For each singular vector we train a logistic regression to predict the spurious label on the training set and then evaluate on the test split. Figure 4 shows these results for all datasets. For models trained on MNIST-CIFAR, the first direction is surprisingly predictive of the MNIST label both for the training set and the test set, independently of which method was used to train. What is remarkable is that this happens even if the spurious label is not actually useful for the task: models trained on datasets with no correlation between the spurious and target label still retain information about the spurious label. This behaviour happens also in more complex datasets like Waterbirds and CelebA, with the first and second directions, respectively being highly predictive of the spurious label. This behavior is consistent across all methods, which suggests that all of them store spurious information in their representations and there is not much of a mechanism to curb this.

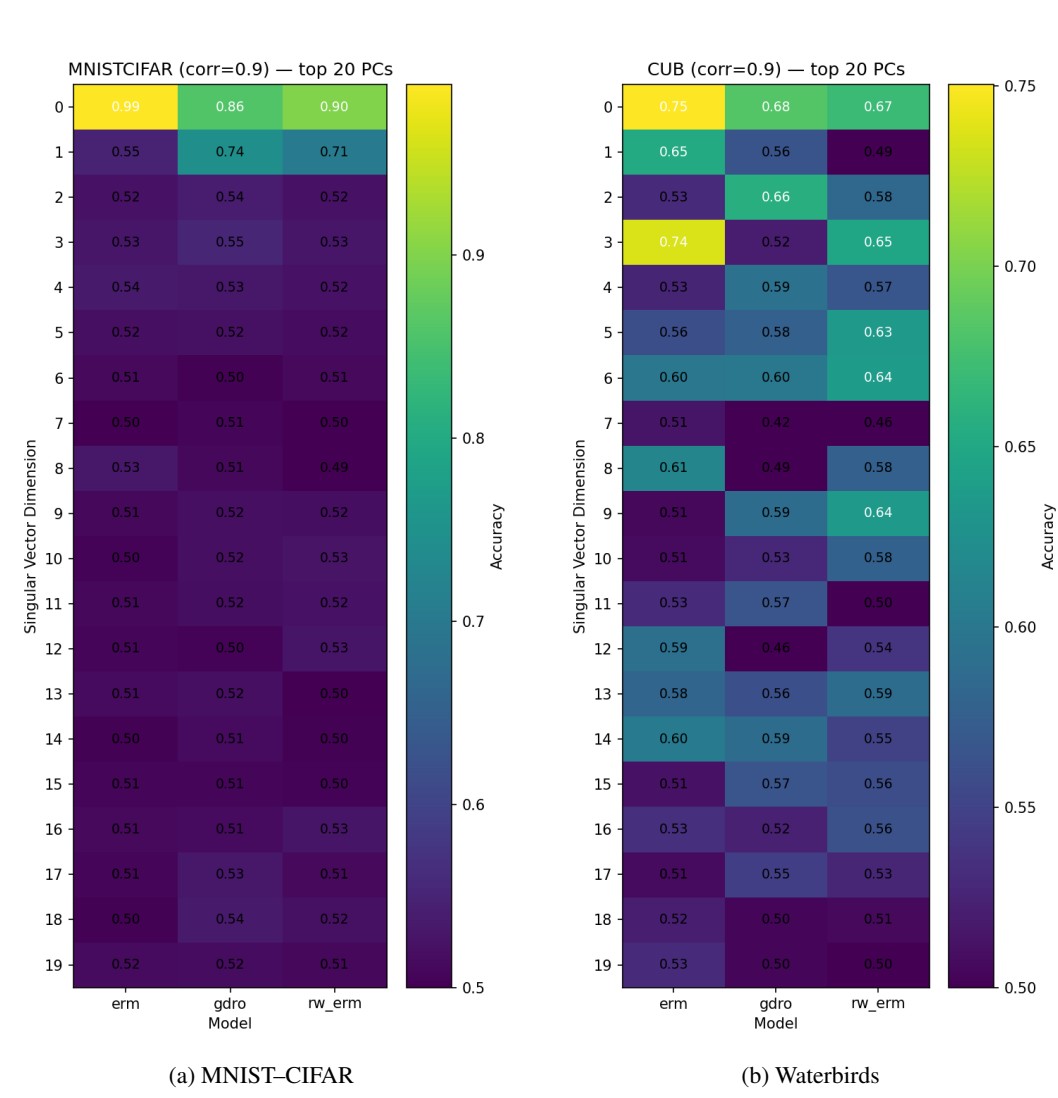

(a) MNIST–CIFAR

(b) Waterbirds

Figure 4: Test accuracy for predicting the spurious label (corr=0.9) across the top-20 singular vectors for models trained on five datasets. For all methods we find highly predictive spurious directions, often within the top-3 singular vectors, suggesting their relevance.

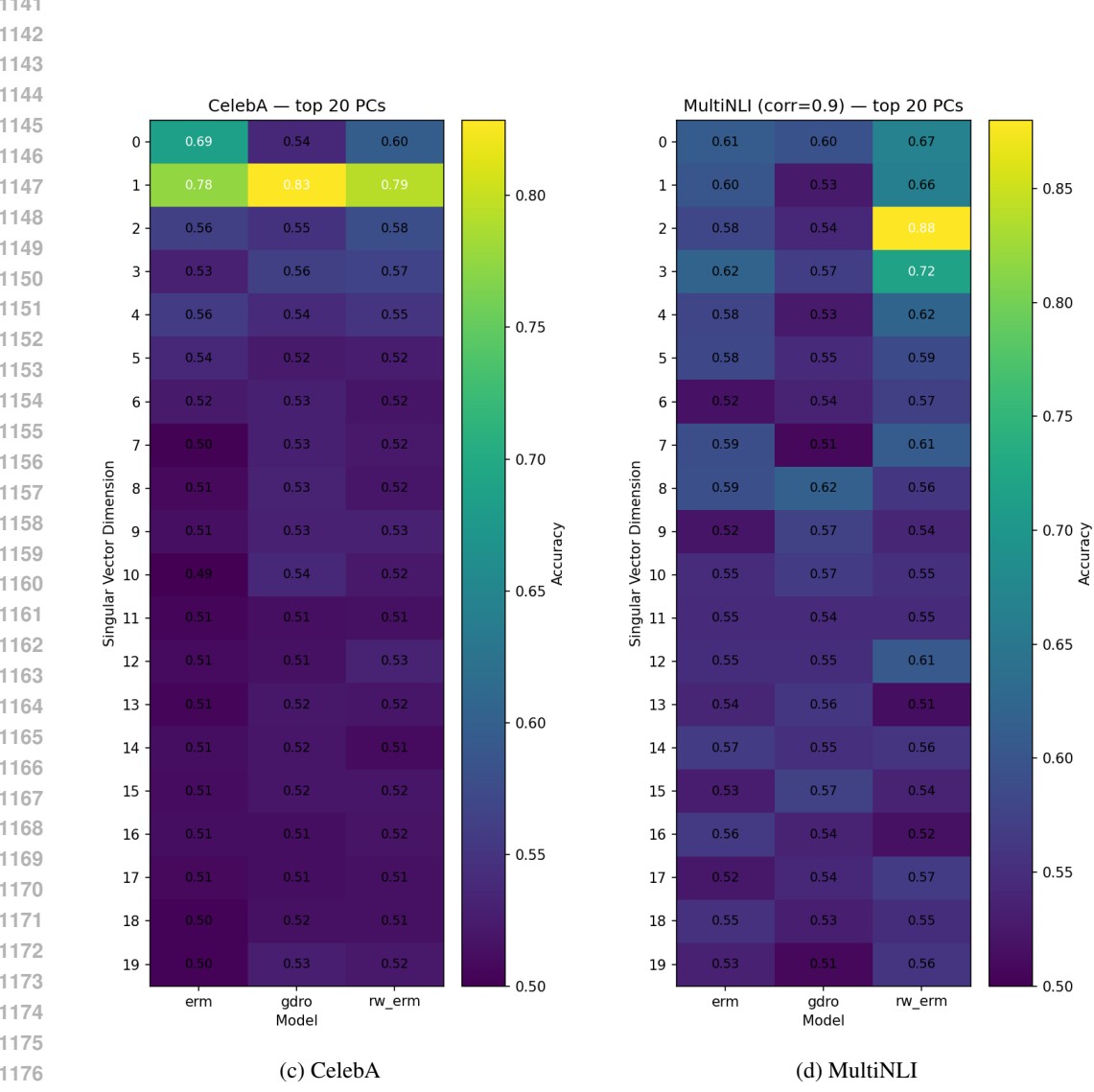

(c) CelebA                         (d) MultiNLI

Figure 4: (continued)

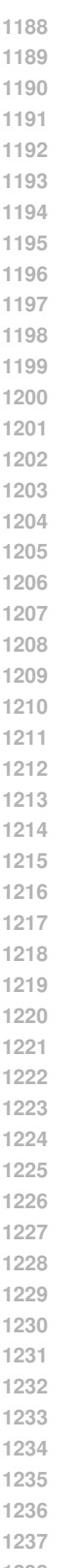
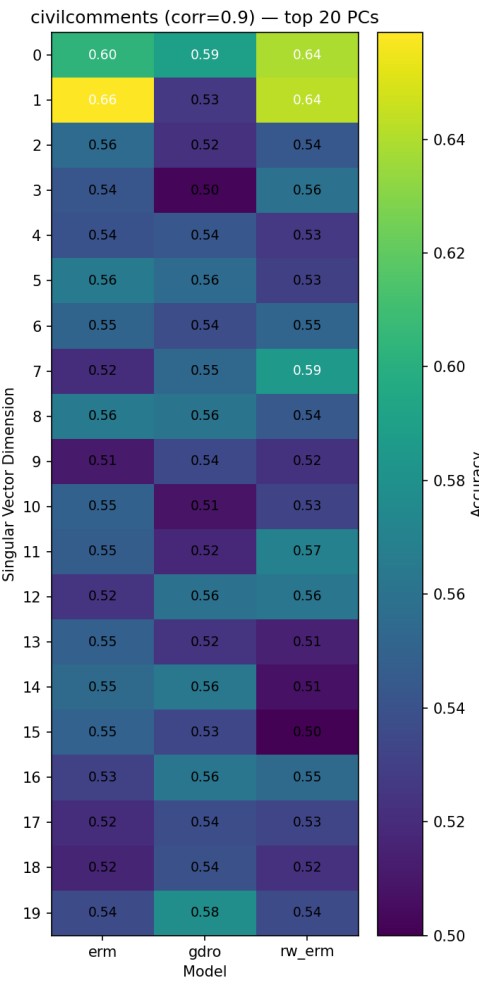

(e) CivilComments

Figure 4: (continued)

# D ADDITIONAL EXPERIMENTS

## D.1 ABLATION ON DATASET SIZE FOR RW+ AND GDRO+

| DATASET | METHOD | 0.1 | 0.25 | 0.5 | 0.75 | 0.9 |
|---|---|---|---|---|---|---|
| MNIST–CIFAR | RW-FT | **80.63%** (1.19) | 79.33% (1.20) | 78.88% (2.79) | 79.28% (2.07) | **80.40%** (0.81) |
| | GDRO-FT | 78.77% (2.54) | 78.53% (1.46) | 78.01% (2.11) | 78.96% (0.75) | 78.22% (1.51) |
| | RW+ | 75.85% (5.13) | 79.79% (1.81) | **80.59%** (1.01) | **80.72%** (0.40) | 79.12% (3.14) |
| | GDRO+ | 76.73% (5.59) | **81.39%** (0.61) | 80.59% (0.23) | 80.19% (0.46) | 79.25% (2.91) |
| WATERBIRDS | RW-FT | 49.43% (1.04) | 56.28% (2.16) | 62.88% (3.74) | 66.20% (2.36) | 66.25% (0.80) |
| | GDRO-FT | 53.32% (1.72) | 62.36% (3.28) | 67.68% (0.70) | 72.95% (3.74) | 71.86% (1.26) |
| | RW+ | 79.22% (7.19) | 83.84% (2.16) | 86.64% (0.65) | 88.00% (0.82) | **88.99%** (0.55) |
| | GDRO+ | **79.85%** (7.42) | **84.33%** (2.66) | **86.75%** (0.82) | **88.21%** (0.78) | 88.92% (0.35) |
| CELEBA | RW-FT | 83.15% (0.64) | 84.63% (1.95) | 85.74% (0.85) | 86.48% (1.70) | 86.85% (1.40) |
| | GDRO-FT | **87.41%** (2.57) | **89.60%** (1.10) | **89.09%** (1.18) | **89.26%** (1.79) | **90.44%** (0.53) |
| | RW+ | 83.70% (0.85) | 84.63% (1.95) | 85.56% (0.56) | 85.00% (0.96) | 85.19% (0.85) |
| | GDRO+ | 80.74% (4.10) | 86.30% (2.10) | 88.15% (0.64) | 88.70% (0.85) | 88.71% (0.85) |

Table 2: Worst Group Accuracy results for various methods trained on all datasets (correlation=0.9) on different percentages of training data. All methods finetuned a classifier from frozen features derived from ERM training. Note that RW+/GDRO+ already uses around 5-10% of the original training data. Even with 50% of their training data, very competitive results are obtained with GDRO+.

## D.2 Explained Variance of PCA for all methods

| Dataset | Correlation | Method | Explained Ratio | Min | Max | PCA dims |
|---|---|---|---|---|---|---|
| Waterbirds | 0.00 | ERM | 91.34% (0.22) | 91.14 | 91.58 | 200 |
| Waterbirds | 0.00 | GDRO | 92.05% (0.95) | 91.37 | 93.13 | 200 |
| Waterbirds | 0.00 | RW | 89.45% (0.16) | 89.31 | 89.63 | 200 |
| Waterbirds | 0.25 | ERM | 90.69% (0.57) | 90.14 | 91.28 | 200 |
| Waterbirds | 0.25 | GDRO | 95.06% (4.26) | 92.14 | 99.95 | 200 |
| Waterbirds | 0.25 | RW | 90.10% (1.08) | 89.40 | 91.35 | 200 |
| Waterbirds | 0.50 | ERM | 91.25% (0.16) | 91.06 | 91.38 | 200 |
| Waterbirds | 0.50 | GDRO | 93.23% (2.62) | 91.40 | 96.23 | 200 |
| Waterbirds | 0.50 | RW | 90.07% (1.02) | 89.32 | 91.23 | 200 |
| Waterbirds | 0.75 | ERM | 91.31% (0.02) | 91.30 | 91.33 | 200 |
| Waterbirds | 0.75 | GDRO | 95.11% (0.28) | 94.89 | 95.42 | 200 |
| Waterbirds | 0.75 | RW | 89.46% (0.05) | 89.42 | 89.52 | 200 |
| Waterbirds | 0.90 | ERM | 90.99% (0.12) | 90.85 | 91.06 | 200 |
| Waterbirds | 0.90 | GDRO | 89.45% (0.43) | 89.08 | 89.92 | 200 |
| Waterbirds | 0.90 | RW | 89.39% (0.06) | 89.32 | 89.44 | 200 |
| CelebA | 0.25 | ERM | 89.21% (0.08) | 89.15 | 89.30 | 200 |
| CelebA | 0.25 | GDRO | 90.24% (0.21) | 90.02 | 90.43 | 200 |
| CelebA | 0.25 | RW | 89.92% (0.26) | 89.69 | 90.20 | 200 |
| MNISTCIFAR | 0.00 | ERM | 87.56% (0.56) | 87.06 | 88.16 | 50 |
| MNISTCIFAR | 0.00 | GDRO | 87.75% (0.72) | 87.22 | 88.56 | 50 |
| MNISTCIFAR | 0.00 | RW | 87.58% (0.64) | 87.00 | 88.26 | 50 |
| MNISTCIFAR | 0.25 | ERM | 88.53% (0.29) | 88.20 | 88.72 | 50 |
| MNISTCIFAR | 0.25 | GDRO | 87.78% (0.63) | 87.11 | 88.37 | 50 |
| MNISTCIFAR | 0.25 | RW | 87.74% (0.65) | 87.07 | 88.37 | 50 |
| MNISTCIFAR | 0.50 | ERM | 89.37% (0.40) | 88.98 | 89.77 | 50 |
| MNISTCIFAR | 0.50 | GDRO | 87.27% (0.54) | 86.75 | 87.82 | 50 |
| MNISTCIFAR | 0.50 | RW | 87.30% (0.62) | 86.77 | 87.99 | 50 |
| MNISTCIFAR | 0.75 | ERM | 90.89% (0.96) | 90.07 | 91.95 | 50 |
| MNISTCIFAR | 0.75 | GDRO | 87.33% (1.07) | 86.36 | 88.48 | 50 |
| MNISTCIFAR | 0.75 | RW | 87.24% (1.10) | 86.20 | 88.39 | 50 |
| MNISTCIFAR | 0.90 | ERM | 92.52% (0.97) | 91.70 | 93.59 | 50 |
| MNISTCIFAR | 0.90 | GDRO | 89.62% (1.40) | 88.13 | 90.90 | 50 |
| MNISTCIFAR | 0.90 | RW | 89.39% (1.27) | 87.96 | 90.39 | 50 |
| MultiNLI | 0.90 | ERM | 99.09% (0.35) | 98.87 | 99.49 | 50 |
| MultiNLI | 0.90 | GDRO | 99.42% (0.13) | 99.28 | 99.52 | 50 |
| MultiNLI | 0.90 | RW | 99.07% (0.21) | 98.92 | 99.32 | 50 |
| CivilComments | 0.90 | ERM | 99.23% (0.18) | 99.03 | 99.33 | 50 |
| CivilComments | 0.90 | GDRO | 99.32% (0.06) | 99.26 | 99.37 | 50 |
| CivilComments | 0.90 | RW | 99.36% (0.10) | 99.25 | 99.44 | 50 |

Table 3: Percentage of Explained Variance for all PCA decompositions used for calculating DCI metrics.

## D.3   DCI METRICS IN FULL

| DATASET | CORR | DISENTANGLEMENT | | | COMPLETENESS | | | INFORMATIVENESS | | |
|---|---|---|---|---|---|---|---|---|---|---|
| | | ERM | GDRO | RW | ERM | GDRO | RW | ERM | GDRO | RW |
| MNIST–CIFAR | 0.00 | 0.992 | 0.991 | 0.993 | 0.768 | 0.765 | 0.769 | 0.957 | 0.957 | 0.957 |
| | 0.25 | 0.545 | 0.449 | 0.454 | 0.714 | 0.697 | 0.700 | 0.955 | 0.957 | 0.957 |
| | 0.50 | 0.252 | 0.230 | 0.231 | 0.757 | 0.752 | 0.752 | 0.958 | 0.963 | 0.963 |
| | 0.75 | 0.106 | 0.105 | 0.105 | 0.841 | 0.840 | 0.838 | 0.968 | 0.973 | 0.974 |
| | 0.90 | 0.030 | 0.016 | 0.017 | 0.921 | 0.819 | 0.865 | 0.984 | 0.983 | 0.984 |
| WATERBIRDS | 0.00 | 0.984 | 0.884 | 0.774 | 0.890 | 0.825 | 0.839 | 0.974 | 0.971 | 0.967 |
| | 0.25 | 0.774 | 0.699 | 0.695 | 0.831 | 0.722 | 0.816 | 0.974 | 0.959 | 0.971 |
| | 0.50 | 0.369 | 0.479 | 0.317 | 0.802 | 0.758 | 0.775 | 0.979 | 0.974 | 0.976 |
| | 0.75 | 0.130 | 0.188 | 0.145 | 0.875 | 0.778 | 0.797 | 0.990 | 0.984 | 0.980 |
| | 0.90 | 0.030 | 0.045 | 0.044 | 0.895 | 0.845 | 0.853 | 0.992 | 0.983 | 0.985 |
| CELEBA | 0.87 | 0.432 | 0.812 | 0.665 | 0.829 | 0.817 | 0.785 | 0.940 | 0.927 | 0.927 |
| MULTINLI | 0.22 | 0.500 | 0.619 | 0.687 | 0.576 | 0.558 | 0.702 | 0.954 | 0.933 | 0.920 |
| CIVILCOMMENTS | 0.52 | 0.649 | 0.716 | 0.593 | 0.622 | 0.531 | 0.584 | 0.881 | 0.858 | 0.882 |

Table 4: Disentaglement, Completeness and Informativenes results for the train set for multiple datasets and methods.

| DATASET | CORR | DISENTANGLEMENT | | | COMPLETENESS | | | INFORMATIVENESS | | |
|---|---|---|---|---|---|---|---|---|---|---|
| | | ERM | GDRO | RW | ERM | GDRO | RW | ERM | GDRO | RW |
| MNIST–CIFAR | 0.00 | 0.992 | 0.991 | 0.993 | 0.768 | 0.765 | 0.769 | 0.939 | 0.942 | 0.942 |
| | 0.25 | 0.545 | 0.449 | 0.454 | 0.714 | 0.697 | 0.700 | 0.926 | 0.930 | 0.928 |
| | 0.50 | 0.252 | 0.230 | 0.231 | 0.757 | 0.752 | 0.752 | 0.873 | 0.888 | 0.886 |
| | 0.75 | 0.106 | 0.105 | 0.105 | 0.841 | 0.840 | 0.838 | 0.755 | 0.798 | 0.801 |
| | 0.90 | 0.030 | 0.016 | 0.017 | 0.921 | 0.819 | 0.865 | 0.622 | 0.647 | 0.651 |
| WATERBIRDS | 0.00 | 0.984 | 0.884 | 0.774 | 0.890 | 0.825 | 0.839 | 0.937 | 0.938 | 0.940 |
| | 0.25 | 0.774 | 0.699 | 0.695 | 0.831 | 0.722 | 0.816 | 0.928 | 0.901 | 0.930 |
| | 0.50 | 0.369 | 0.479 | 0.317 | 0.802 | 0.758 | 0.775 | 0.921 | 0.913 | 0.915 |
| | 0.75 | 0.130 | 0.188 | 0.145 | 0.875 | 0.778 | 0.797 | 0.897 | 0.877 | 0.882 |
| | 0.90 | 0.030 | 0.045 | 0.044 | 0.895 | 0.845 | 0.853 | 0.866 | 0.815 | 0.819 |
| CELEBA | 0.87 | 0.432 | 0.812 | 0.665 | 0.829 | 0.817 | 0.785 | 0.935 | 0.925 | 0.925 |
| MULTINLI | 0.22 | 0.500 | 0.619 | 0.687 | 0.576 | 0.558 | 0.702 | 0.886 | 0.879 | 0.892 |
| CIVILCOMMENTS | 0.52 | 0.649 | 0.716 | 0.593 | 0.622 | 0.531 | 0.584 | 0.868 | 0.849 | 0.865 |

Table 5: Disentaglement, Completeness and Informativenes results for the test set for multiple datasets and methods.

# E    LIMITATIONS AND SOCIETAL IMPACT

Our analysis hinges on the following assumptions: groups within the dataset are usually unbalanced with the most represented groups benefiting from the spurious correlation. However, this is a standard setting in the robustness literature. Our theoretical analysis depends on a specific definition of a spurious vector, with which some may disagree.

Our proposed method requires extra data with annotations of both the class and spurious label for the finetuning stage. It does not require much of it, 5% for the full method, but Table 3 shows decent results with even less. All competing baselines require at least this much extra data.

We believe this work may have a positive societal impact as it is trying to mitigate a problem of spurious correlations that usually happens on underrepresented subsets of the data, which may represent in practice underrepresented parts of society. Moreover, our work seeks to understand the mechanisms by which GRMs work which could aid in developing better methods that work to achieve robustness and fairness.

# F EXPERIMENT DETAILS

## F.1 HYPERPARAMETERS

### F.1.1 GENERAL

All methods use SGD with momentum 0.9 as their optimizer for vision datasets. For text datasets we used AdamW. It is the same setup as Liu et al. (2021). We use L2 regularization and no data augmentation. No learning rate scheduler is used. All methods are run for 3 seeds. Seeds used were: $\{111, 222, 333\}$ Learning rates and weight decay used were taken from (Liu et al., 2021), which were based off of (Sagawa* et al., 2020).

### F.1.2 MNIST-CIFAR

We train all models for 5000 epochs/iterations. L2 regularization of $10^{-4}$, learning rate of 0.001 for all methods. Batch size is 10000.

### F.1.3 WATERBIRDS

We train all models for 300 epochs. Batch size is 64. For ERM, we use L2 regularization of $10^{-4}$, learning rate of $10^{-4}$; for GDRO L2 regularization of 1, learning rate of $10^{-5}$; for RW, L2 regularization of $10^{-3}$, learning rate of $10^{-4}$.

### F.1.4 CELEBA

We train all models for 50 epochs. Batch size is 64. For ERM, we use L2 regularization of $10^{-4}$, learning rate of $10^{-4}$; for GDRO L2 regularization of 0.1, learning rate of $10^{-5}$; for RW, L2 regularization of 0.1, learning rate of $10^{-5}$.

### F.1.5 MULTINLI

We train all models for 50 epochs. Batch size is 64. For ERM, we use L2 regularization of $10^{-4}$, learning rate of $10^{-4}$; for GDRO L2 regularization of 0.1, learning rate of $10^{-5}$; for RW, L2 regularization of 0.1, learning rate of $10^{-5}$.

### F.1.6 CIVILCOMMENTS

We train all models for 50 epochs. Batch size is 64. For ERM, we use L2 regularization of $10^{-4}$, learning rate of $10^{-4}$; for GDRO L2 regularization of 0.1, learning rate of $10^{-5}$; for RW, L2 regularization of 0.1, learning rate of $10^{-5}$.

# G CO2 EMISSION RELATED TO EXPERIMENTS

Experiments were conducted using a private infrastructure, which has a carbon efficiency of 0.432 $kgCO_2eq/kWh$. A cumulative of 2275 hours of computation was performed on hardware of type GTX 1080 Ti (TDP of 250W).

Total emissions are estimated to be 245.7 $kgCO_2eq$ of which 0 percents were directly offset.

Estimations were conducted using the MachineLearning Impact calculator presented in Lacoste et al. (2019).

# H LLM USAGE

LLMs were only used in polishing writing and in aiding in creating tables and plots.

