# OpenReview forum: "Alignment, Convexity and Completeness: Mechanisms Behind GroupDRO"
_ICLR.cc/2026/Conference — Submitted to ICLR 2026_

### Official Review · Reviewer_cURQ · 2025-10-28

**Soundness:** 2
**Presentation:** 1
**Contribution:** 1
**Rating:** 2
**Confidence:** 4

**Summary:**

This paper compares the classifier learned by GroupDRO, which minimizes the maximum loss on any subpopulation in the dataset, with the standard ERM classifier. Theoretical analysis is provided for a linear model with fixed features on a four-points dataset, and it is shown that the GroupDRO classifier aligns more with the non-spurious classifier. Experiments are also provided which analyze the effect of GroupDRO on the neural network representation; it is hypothesized that GroupDRO performs better because it distributes weight across multiple different features instead of focusing its magnitude on a single shortcut.

**Strengths:**

The idea to use the DCI framework [1] for analysis of neural network representations under spurious correlations is interesting (though insufficiently explored, see Weakness 4).

[1] Eastwood and Williams.  A Framework for the Quantitative Evaluation of Disentangled Representations. ICLR 2018.

**Weaknesses:**

1. It is unclear whether this paper studies the standard GroupDRO formulation, i.e., minimizing the maximum loss over any group. In Section 2.2.3 the paper defines GroupDRO as a version of reweighting with group-wise softmax weights. Minimizing the loss $\mathcal{L}^{DRO}$ is equivalent to minimizing the standard GroupDRO loss only as $\epsilon \to \infty$, a regime which is not discussed. While the theoretical results appear to use the typical min-max definition, the GroupDRO algorithm used for the empirical results in Section 4 is not made explicit, and it may differ from results obtained via the min-max objective. See [1, Section 4] for further discussion of the relationship between GroupDRO and importance weighting.

2. The second listed contribution, “Role of regularization”, is not novel. The fact that adding $\ell_2$ regularization to a convex loss induces strong convexity is one of its most basic properties. I also believe Theorem B.1, the main technical result in this paper upon which Proposition B.6 follows by definitions of ERM and strong convexity, is not very novel. The main observation of Theorem B.1 is that for convex losses the optimal GroupDRO classifier corresponds to an importance-weighted classifier with certain weights, but this is exactly the characterization of Proposition 1 of [1].

3. From a technical perspective, the theoretical results are not exceptionally novel or sophisticated, and they do not introduce any techniques which might be generalizable beyond the scope of this paper. For example, analysis of linear models with fixed features on the four-points dataset is a simple low-dimensional setting which has been well-studied [2]. Moreover, no theoretical analysis of representation learning is provided despite this having been identified as an important challenge in the literature [3] and addressed in the experiments. Finally, the theoretical results study the optimal GroupDRO classifier $\theta^*$ and not the actual SGD solution; it is unclear whether gradient descent/flow will converge to the min-max solution, i.e., under the continuous-time alternating minimization-like gradient flow of Prop. B.4.

4. The empirical study is not convincing and overall lacking in rigor:

    a. Section 4.1 and the associated Table 1 do not provide additional insights beyond similar experiments already presented in [3, 8, 9].

    b. The metrics of the DCI framework are not formally defined, and how they are computed is not specified. In addition, metrics are computed after PCA reduction to only 50 dimensions, and the amount of preserved variance is not quantified, nor are comparisons made between metrics computed on PCA and non-PCA features.

    c. The claim that “completeness is consistently lower for GDRO” is insufficiently supported. First, the strength of the claim is limited as GroupDRO matches the completeness of other methods in MNIST-CIFAR and CelebA, and is lowest by only a small amount on MultiNLI and 4/5 of the correlation settings on Waterbirds. Second, error bars are not provided in Figure 2, making it difficult to assess whether the lower completeness may be due to randomness or hyperparameters.

5. Finally, comparison to previous work in theoretical analysis of spurious correlations and GroupDRO is severely lacking. Here are a few references within which the theoretical contributions of this paper should be contextualized: [2, 4, 5, 6, 7]

[1] Sagawa et al. Distributionally Robust Neural Networks for Group Shifts: On the Importance of Regularization for Worst-Case Generalization. ICLR 2020.

[2] Nagarajan et al. Understanding the failure modes of out-of-distribution generalization. ICLR 2021.

[3] Izmailov et al. On Feature Learning in the Presence of Spurious Correlations. NeurIPS 2022.

[4] Wang and Wang. On the Effect of Key Factors in Spurious Correlation: A Theoretical Perspective. AISTATS 2024.

[5] Puli et al. Don't blame Dataset Shift! Shortcut Learning due to Gradients and Cross Entropy. NeurIPS 2023.

[6] Sagawa et al. An Investigation of Why Overparameterization Exacerbates Spurious Correlations. ICML 2020.

[7] Holstege et al. Optimizing importance weighting in the presence of sub-population shifts. ICLR 2025.

[8] Kirichenko et al. Last Layer Re-Training is Sufficient for Robustness to Spurious Correlations. ICLR 2023.

[9] Idrissi et al. Simple data balancing achieves competitive worst-group-accuracy. CLeaR 2022.

**Questions:**

Please see the Weaknesses section; I would especially appreciate clarifications on points (1) and (4).

---

> ### Author Response · Authors · 2025-11-18
> **Response 1/2**
>
> We thank the reviewer for their time in reviewing our work. We will proceed to address the weaknesses listed in your review:
>
> **W1: "It is unclear whether this paper studies the standard GroupDRO formulation..."**
>
> **R**: In our theoretical analysis, we study the standard min–max formulation of GDRO; we added a line to the paper in Section 3 indicating we are using the Min-Max formulation for that analysis. For our experiments, we use the reweighting implementation described in the text, which is the practical variant adopted by works using the codebase of [1]. We have added text indicating we use that formulation for experiments when GDRO is described. We agree that the two formulations may behave differently in practice, but our goal is to understand the mechanisms that make GDRO work well empirically, using the theory as a guide for what to look for. Notably, we find that the empirical behavior of the reweighting implementation aligns with the predictions from our min–max analysis, which is itself a nontrivial and interesting observation.
>
> **W2: "The second listed contribution, “Role of regularization”, is not novel."**
>
> **R:** We believe this is poor wording of the second contribution. Theorem B.1 is not the main technical result of this work, which is why it's listed under the "Preliminaries" subsection of the "Derivation of Theoretical Results" in the Appendix. As stated by Reviewer cv7c, the spirit of the contribution is to theoretically **"explain why strong L2 regularization improves GDRO, which had been previously empirical folklore"**. The second contribution should state *"Using the earlier link between alignment and performance for μ-strong losses, we show that GDRO needs L2 regularization on the (non-μ-strong) Cross-Entropy loss because this regularization makes the loss μ-strong."* We have rewritten the second contribution to reflect this.
>
> **W3: From a technical perspective, the theoretical results are not exceptionally novel or sophisticated...**
>
> **R:** The analysis of fixed-feature classifiers is relevant because, following DFR, many recent methods use fixed ERM features and only finetune the classifier (as discussed in Related Work). We agree that a full theoretical treatment of representation learning for GDRO is challenging and important. Our work contributes toward this goal in two ways: (a) we show that reducing alignment with spurious directions is beneficial, and (b) we show that lower completeness of the representation helps when classifier alignment alone is insufficient. These two insights provide a starting point for theory on representation learning under GRMs, raising questions such as: *How can models structure features so that spurious components are easier for the classifier to ignore? How can models induce lower completeness in their learned features?*
>
> **W4: "The empirical study is not convincing and overall lacking in rigor:**
>
> **R:** Section 4.1 isolates the effect of finetuning only the classifier with GDRO to assess the importance of classifier alignment. We show that this alignment effect is strong but does not fully explain GDRO’s performance gains (e.g., on Waterbirds and MNIST–CIFAR 0.9), which motivates searching for an additional mechanism. This leads to our finding that GDRO also benefits from inducing lower completeness in the representations when alignment alone is insufficient. Prior works [3, 8, 9] do not report the specific comparisons or datasets needed for this analysis: they do not compare full GDRO against different finetuning strategies (GDRO, RW, or pure subsampling), nor do they provide multiple MNIST–CIFAR and Waterbirds variants with varying correlation strengths to study how increasing correlation affects performance.
>
> **b. The metrics of the DCI framework are not formally defined..."**
>
> **R:**  In the paper we describe DCI only at a high level; for formal definitions we refer the reviewer to the original DCI paper, whose official implementation we use to compute all our metrics. We initially used a reduced set of features for PCA because DCI is computationally expensive on high-dimensional representations. However, we have now recomputed the PCA-based DCI scores for CelebA and Waterbirds using all correlations with  N=200 dimensions (≈60% explained variance for these datasets) and confirmed that the results do not change. We have updated the paper so that all methods use PCA configurations with ≈90% or more explained variance (often >99%), and we added an appendix section (D.2) detailing these variances. We do not compare PCA vs. non-PCA features because: (a) the original, non-PCA representations are very high-dimensional and prohibitively expensive for DCI, and (b) PCA and non-PCA features differ only by a linear transformation that the classifier can learn. Given this and the high explained variance of our PCA projections, we believe an explicit comparison is unnecessary.

---

> > ### Author Response · Authors · 2025-11-18
> > **Response 2/2**
> >
> > **c. The claim that “completeness is consistently lower for GDRO” is insufficiently supported. First, the strength of the claim is limited as GroupDRO matches the completeness of other methods in MNIST-CIFAR and CelebA, and is lowest by only a small amount on MultiNLI and 4/5 of the correlation settings on Waterbirds. Second, error bars are not provided in Figure 2, making it difficult to assess whether the lower completeness may be due to randomness or hyperparameters.**
> >
> > **R:**
> > * Our claim is specifically about GDRO vs. ERM, not RW. On Waterbirds, GDRO has consistently lower completeness than ERM for all correlation levels (ERM is the upper green curve), and GDRO’s completeness is lower on average than ERM’s across all datasets.
> > * We have now added error bars to Figure 2 to show variability across runs. In the few cases where GDRO and ERM have overlapping error bars, our interpretation is as follows: our paper argues that GDRO can exploit two mechanisms—(i) reduced alignment with spurious directions and (ii) lower completeness. In many settings, the alignment effect alone suffices (MNIST–CIFAR 0.0–0.75, CelebA, MultiNLI, CivilComments). When alignment is not enough (e.g., all Waterbirds settings and MNIST–CIFAR 0.9), GDRO also leverages lower completeness. This is visible by comparing GDRO vs. GDRO-FT performance and the corresponding ERM vs. GDRO completeness gap in those regimes.
> >
> > **W5: Finally, comparison to previous work in theoretical analysis of spurious correlations and GroupDRO is severely lacking. Here are a few references within which the theoretical contributions of this paper should be contextualized: [2, 4, 5, 6, 7]**
> >
> > R: We thank the reviewer for these references. These works have largely focused on (i) identifying properties of the training data that cause models to fail under spurious correlations or (ii) explaining why ERM fails in such settings. By contrast, our goal is to analyze how GDRO itself behaves—both at the classifier level and in how it shapes representations—given that GDRO is known to perform well under spurious correlations. Our analysis makes relatively mild assumptions on the data, beyond the existence of minority groups.
> > In particular, [2], [4], and [6] characterize data properties that induce failures under spurious correlations but do not explain how GDRO succeeds; [5] studies ERM’s inductive bias toward max-margin solutions but does not address GDRO; [7] highlights issues with importance weighting (including in GDRO) under finite samples, but does not analyze the mechanisms by which GDRO achieves robustness. For these reasons, and due to page limits, we did not originally include these works. However, we agree that explicitly positioning our contributions relative to them would improve the paper, and we will gladly add a discussion of [2, 4, 5, 6, 7] to the Related Work section when the page limit is increased.

---

> > > ### Comment · Reviewer_cURQ · 2025-11-26
> > >
> > > I appreciate the authors' responses to each of my points. I believe that the paper will be improved by integrating further clarification and elaboration on each of the authors' explanations. Nevertheless, my main concerns about the GroupDRO formulation, novelty of key theoretical results/techniques, and rigor of the experiments still remain. For these reasons, I will retain my recommendation.

---

### Official Review · Reviewer_cv7c · 2025-10-31

**Soundness:** 3
**Presentation:** 3
**Contribution:** 3
**Rating:** 8
**Confidence:** 2

**Summary:**

This paper presents a theoretically grounded and empirically validated analysis of why GDRO outperforms ERM under spurious correlations. It shows that GDRO learns classifiers that align less with spurious directions and more with non-spurious directions, and that under µ-strongly convex losses, this alignment gap directly bounds the worst-group performance difference between ERM and GDRO. The authors further demonstrate that adding L2 regularization induces strong convexity, offering a principled explanation for GDRO’s reliance on strong L2 penalties. Extensive experiments on both image and text benchmarks confirm the theoretical predictions and reveal that GDRO not only adjusts the classifier head but also reshapes the learned representation by lowering completeness.

**Strengths:**

Alignment-based reasoning reflected in consistent empirical behavior across multiple benchmarks in both vision and language domains.

Explains why strong L2 regularization improves GDRO, which had been previously empirical folklore.

The paper is clearly written, well-organized, and supported by reproducible experiments.

**Weaknesses:**

The theoretical framework is largely developed under the fixed-feature linear classifier assumption (Section 3.1 and Appendix B).

Several simplifying assumptions, including symmetric group shifts, shared subspaces, and identical loss functions overlook real-world heterogeneity.

The completeness analysis in Section 5.3 focuses on correlations rather than causal relationships, leaving it uncertain whether reduced completeness plays a direct role in enhancing robustness.

Clarifying these aspects or discussing their implications would strengthen the overall contribution.

**Questions:**

See weakness

---

> ### Author Response · Authors · 2025-11-18
> **Response to Reviewer cv7c**
>
> We thank the reviewer for their time and for showing enthusiasm and appreciation for our work and describing it as *“clearly written, well-organized, and supported by reproducible experiments”*. We proceed to address the weakness and questions:
>
> **C1: "The theoretical framework is largely developed under the fixed-feature linear classifier assumption (Section 3.1 and Appendix B).”**
>
> **R:** We agree that this is an idealized setting, but we believe it is still highly relevant. Many recent robustness methods starting with DFR use fixed ERM features and only finetune a linear classifier, and our analysis is designed to match this regime. Our results show that the alignment effect induced by finetuning the classifier with GDRO is already quite strong, but also that it is not sufficient to fully explain GDRO’s gains—this is precisely where differences in representation completeness start to matter.
>
> **C2: “Several simplifying assumptions, including symmetric group shifts, shared subspaces, and identical loss functions overlook real-world heterogeneity.”**
>
> **R:** These assumptions are made to keep the theoretical analysis tractable. To compensate, we explicitly evaluate GDRO and ERM on multiple datasets of increasing real-world complexity, and we observe that the qualitative predictions of our theory remain consistent even when the assumptions are only approximate. We would also appreciate clarification on what is meant by “identical loss functions” in this context (e.g., whether this refers to the same loss form across groups, or something more specific), so that we can address this point more precisely in the revision.
>
> **C3: “The completeness analysis in Section 5.3 focuses on correlations rather than causal relationships, leaving it uncertain whether reduced completeness plays a direct role in enhancing robustness.”**
>
> **R**: We agree that our analysis is correlational: we do not claim to establish a causal effect of reduced completeness on robustness. Testing such a causal link would require a way to systematically intervene on completeness in the representation, which is beyond the scope of this work. Our contribution here is to document a consistent empirical pattern: when GDRO is used instead of standard ERM, we observe lower completeness in the learned representations in precisely those regimes where classifier alignment alone does not explain GDRO’s robustness. We view this as a hypothesis-generating result that future work on causal mechanisms could build on.

---

### Official Review · Reviewer_CjL4 · 2025-11-05

**Soundness:** 1
**Presentation:** 1
**Contribution:** 2
**Rating:** 2
**Confidence:** 2

**Summary:**

Objective of the paper is to theoretically explain the merits of Group-DRO (sagawa et.al.) and motivate the need for strong parameter regularization in Group-DRO.

**Strengths:**

Sub-population shift is a problem with practical significance.

**Weaknesses:**

The main contribution of the paper is theoretical analysis, but I find the corresponding section 3 very hard to parse to the best of my efforts. It contained either too many symbols or symbols that are not defined. Too many to even list them out here. I expected the authors would present a summary (in vernacular text) before or after proposition statement, but they did not. In any case, what the authors embark to prove in section 3 is too simplified, classifier is trained with frozen representations.

The paper argues that GDRO leads to improved disentanglement of features, which in turn makes it easier for classifier to ignore the spurious features. But from Figure 2 (top row), GDRO is not significantly better than ERM on disentanglement.

**Questions:**

GDRO paper justifies regularization with a simple reason: to avoid overfitting on the minority group. I do not understand the mu-strongly convex argument of the paper but I find the justification of the GDRO paper straightforward. Please explain why it is important to understand regularization beyond the simple argument?

Also, please address the concerns raised in the "weaknesses" section.

---

> ### Author Response · Authors · 2025-11-18
> **Response 1/2**
>
> **C1: The main contribution of the paper is theoretical analysis, but I find the corresponding section 3 very hard to parse to the best of my efforts. It contained either too many symbols or symbols that are not defined. Too many to even list them out here. I expected the authors would present a summary (in vernacular text) before or after proposition statement, but they did not.**
>
> **R:** The structure of the paper is as follows: Section 2 introduces the preliminaries and notation which is critical to understanding Section 3, and Section 3 summarizes the main theoretical results, with full proofs deferred to the appendix. Regarding the comment about undefined symbols, we would be very grateful if the reviewer could indicate a few specific examples (e.g., a particular equation or symbol whose definition is hard to find), so that we can address these points directly. This targeted feedback would help us improve the exposition further, especially in light of another reviewer’s comment that the paper is “clearly written, well-organised.”
> In addition, the current version already includes several informal “takeaway” passages intended to give a verbal summary of the key insights. For example:
> * In Section 3.1, we explicitly state in words that our first results show that GDRO reduces alignment with the spurious direction and increases alignment with the non-spurious direction.
>
> * Just before Section 3.2, we explain that Propositions B.4 and B.5 show that these alignment relations hold along the optimization trajectory, and that the next result links differences in alignment to differences in worst-group performance.
>
> * Around Proposition B.6, we include a full paragraph in prose explaining that GDRO must have smaller alignment with the spurious direction than ERM to obtain lower worst-group loss, that this highlights alignment as a key mechanism, and that adding an L2 penalty makes the loss strongly convex, allowing us to relate $\lambda$, alignments, and losses.
>
>
> We would welcome feedback on how to make these takeaways clearer.
>
>
> **C2: In any case, what the authors embark to prove in section 3 is too simplified, classifier is trained with frozen representations.**
>
> **R:** As stated in the literature on the latest methods for Robustness from Deep Feature Reweighting (DFR) and beyond, most of these methods rely on frozen ERM features and finetune the classifier, indicating that the classifier is the one that struggles the most with spurious correlations. The setting used in this paper follows this line of work, aiming to propose an explanation for the phenomenon in question. However, we take a step forward in this assumption and also analyse the representations, finding one measure in the representation that is induced by GDRO: lower completeness.
>
> **C3: The paper argues that GDRO leads to improved disentanglement of features, which in turn makes it easier for classifier to ignore the spurious features. But from Figure 2 (top row), GDRO is not significantly better than ERM on disentanglement.**
>
> **R:** Our work does *not* claim that GDRO improves disentanglement, nor is this listed among our contributions. We would ask the reviewer to point out where the article argues for this point, as we are unable to find it. In Section 4.2, we explicitly frame disentanglement as a hypothesis about GDRO’s effect on features — *“GDRO produces more disentangled features, making it easier for the classifier to rely on predictive rather than spurious attributes”* — and then test this empirically. Our experiments do not support this hypothesis (or at least, they do not provide sufficient evidence to endorse it). In turn, however, we do find that GDRO does produce consistent lower completeness than ERM, which was unexpected.
> The only place where disentanglement might be read as a positive claim is the caption of Figure 2: “*GDRO frequently increases disentanglement, though the effect is not uniform across datasets…*” This is intended as a direct, descriptive reading of the plotted values, not as a core argument of the paper.
> Our main message is captured by the title, “*Alignment, Completeness and Convexity…*”: these are the mechanisms we identify as key to GDRO’s success—not disentanglement.

---

> > ### Author Response · Authors · 2025-11-18
> > **Response 2/2**
> >
> > **C4: GDRO paper justifies regularization with a simple reason: to avoid overfitting on the minority group. I do not understand the mu-strongly convex argument of the paper but I find the justification of the GDRO paper straightforward. Please explain why it is important to understand regularization beyond the simple argument?"**
> >
> > We agree that the original GDRO paper [1] provides a clear and intuitive empirical justification for L2 regularization (avoiding overfitting on minority groups), and its importance is reflected already in the title of that work. However, that justification is purely empirical: it does not explain theoretically why L2 is necessary for GDRO, nor how it interacts with the mechanism we study (alignment with spurious features).
> >
> > One of our contributions is precisely to fill this gap. We show that adding L2 regularization makes the loss μ-strongly convex, which is a key assumption for our theoretical results that connect (i) differences in alignment with spurious directions to (ii) differences in worst-group performance between GDRO and ERM. In other words, strong convexity provided by L2 is what allows us to rigorously link alignment and performance and to explain why GDRO with L2 behaves better than ERM in our setting.
> > Thus, our goal is not to contradict the original intuitive explanation, but to complement it with a formal account: the usual “avoid overfitting” argument tells us that regularization helps, while our analysis explains how and under which conditions L2 regularization is mathematically tied to GDRO’s robustness via μ-strong convexity.
> >
> > [1] Shiori Sagawa*, Pang Wei Koh*, Tatsunori B. Hashimoto, and Percy Liang. Distributionally robust neural networks for group shifts: On the importance of regularization for worst-case generalization. In International Conference on Learning Representations, 2020. URL https://openreview.net/forum?id=ryxGuJrFvS.

---

### Author Response · Authors · 2025-12-02
**Summary of Discussion so far**

Dear Area Chair,

To facilitate your assessment, we summarize the main concerns raised by the reviewers and how we addressed them in the rebuttal and revised manuscript.

**Reviewer CjL4.**
CjL4 raises three points: (1) difficulty following Section 3 due to perceived lack of notation and intuitive (“vernacular”) explanations; (2) concern that the frozen-representation setting is too simple; and (3) the impression that we claim GDRO improves disentanglement, allegedly not supported by the data.

**Our response.**
*(1) Clarity of the theory section:* This contrasts with reviewer cv7c, who explicitly praises the clarity of the paper, and reviewer cURQ, who engages deeply and positively with the theory. Section 2 already introduces notation, and in the rebuttal and revised manuscript we further clarify Section 3 and add more intuitive explanations of the main results.
*(2) Relevance of frozen representations:* We argue this regime is highly relevant, as many recent robustness methods (e.g., DFR-style approaches) freeze ERM features and fine-tune only the classifier. Our analysis directly targets this actively used setting.
*(3) Claim about disentanglement*: We do not present “GDRO improves disentanglement” as a contribution. It was a hypothesis we explicitly test and find not supported by the data, and we state this negative result clearly. Our main contributions do not rely on GDRO improving disentanglement.

**Reviewer cv7c.**
cv7c raises three concerns: (1) focus on the frozen-representation regime; (2) several simplifying assumptions in the theory; and (3) the completeness analysis being correlational rather than causal.

**Our response**.
(1) Frozen representations: As above, this regime closely matches a widely used practical paradigm (frozen ERM features + linear classifier fine-tuning), making it both realistic and timely.

(2) Simplifying assumptions: Assumptions are introduced to keep the analysis tractable however they are in line with other work. To compensate, we evaluate ERM and GDRO on multiple datasets of increasing real-world complexity, and observe that the qualitative predictions of our theory remain consistent even when these assumptions hold only approximately.

(3) Correlational completeness analysis: We agree that our completeness analysis is correlational and we do not claim a causal effect of reduced completeness on robustness. Establishing such a link would require controlled interventions on completeness, which we consider important but beyond the scope of this work.

**Reviewer cURQ**.
cURQ raises three issues: (1) a discrepancy between the theoretical and empirical formulations of GroupDRO; (2) the novelty of the key theoretical results/techniques; and (3) the rigor of the experiments.

**Our response.**

*(1) Theoretical vs. empirical GroupDRO:* For tractability, our theory uses a min–max formulation of GDRO, while our experiments use the standard formulation of Sagawa et al. Our goal is to understand the effects of GDRO in practice, with the theory serving as a guide. Despite the differing formulations, we observe strong qualitative agreement between theoretical predictions and empirical behavior, which we view as interesting in its own right.

*(2) Novelty of the theoretical contribution:* We believe reviewer cURQ partly misinterpreted our second main contribution, which we have now rewritten for clarity. We do not claim as novel that adding L2 regularization to cross-entropy yields μ-strong convexity—this is a standard, preliminary fact in our analysis. Rather, our contribution is to show how this observation, combined with our first contribution, yields a theoretical explanation for why GDRO requires L2 regularization—an empirical phenomenon repeatedly observed but, to our knowledge, not previously explained.

*(3) Experimental rigor:* The reviewer noted the absence of error bars, insufficiently explained variance in PCA, and questioned whether lower completeness was supported empirically. In the revision, we add error bars across runs and recompute PCA decompositions to retain ≈90% (or more) of the variance; the conclusions remain unchanged. For completeness, the concern seems to stem from comparing Reweighting vs. GDRO in Figure 2. The correct comparison for our claim is ERM vs. GDRO, under which GDRO consistently achieves lower completeness, aligning with our theoretical predictions.

We hope this summary clarifies that the main concerns have been carefully addressed in the rebuttal and revised manuscript, and that overall the paper is clear, tackles a timely and practically relevant setting, and provides both theoretical insight and empirical evidence on the mechanisms by which GDRO achieves group robustness.

Please see the rebuttal for more details.

Sincerely,
The authors

---

### Meta-Review · Area_Chair_8wnA · 2025-12-30

**Summary:**

The paper theoretically analyzes Group Distributional Robust Optimization and explain its success as well as success of L2 regularization using the lens of optimization, convexity and feature completeness.

Regarding reviewer concerns, please see below.

**Reviewer Concerns:**

Concerns addressed by the rebuttal:
- Clarifications regarding GDRO and DCI formulations.
- Restrictive assumptions in the theorems (the authors argued they need to make *some* assumptions, the assumptions are reasonable / common in the literature and empirically observations hold).


Outstanding concerns:
- Novelty of their findings given prior works.
- Motivation for the work: while there is certain value to add additional rigor to existing explanations of empirically successful methods, the novel insights are limited.

**Reviewer Scores:**

Reviewer CjL4: I believe that this reviewer would maintain their score of 2 (or pretty unlikely but could increase to 4) since the rebuttal could only partially resolve their concerns; the clarity concerns need to be fundamentally addressed, while I believe motivation for this work in the light of Sagawa et al work wasn't fully convincing.

Reviewer cv7c: I believe the reviewer would maintain their score 8.

Reviewer cURQ: the reviewer clearly indicated in their response that they maintain their score 2.

I believe final scores would be 2, 2, 8. Given the scores and the detailed feedback and concerns regarding novelty and clarity, I recommend a reject.

---

### Decision · Program_Chairs · 2026-01-26

Reject